# ExPLoRA: Parameter-Efficient Extended Pre-Training to Adapt Vision Transformers under Domain Shifts

**Samar Khanna** [1]  **Medhanie Irgau** [1]  **David B. Lobell** [1]  **Stefano Ermon** [1,2]

## Abstract

Parameter-efficient fine-tuning (PEFT) techniques such as low-rank adaptation (LoRA) can effectively adapt large pre-trained foundation models to downstream tasks using only a small fraction (0.1%-10%) of the original trainable weights. An under-explored question of PEFT is in extending the pre-training phase without supervised labels; that is, can we adapt a pre-trained foundation model to a new domain via efficient self-supervised pre-training on this domain? In this work, we introduce ExPLoRA, a highly effective technique to improve transfer learning of pre-trained vision transformers (ViTs) under domain shifts. Initializing a ViT with pre-trained weights on large, natural-image datasets such as from DinoV2 or MAE, ExPLoRA continues the unsupervised pre-training objective on a new domain, unfreezing 1-2 pre-trained ViT blocks and tuning all other layers with LoRA. We then fine-tune the resulting model only with LoRA on this new domain for supervised learning. Our experiments demonstrate state-of-the-art results on satellite imagery, even outperforming fully pre-training and fine-tuning ViTs. Using the DinoV2 training objective, we demonstrate up to 8% improvement in linear probing top-1 accuracy on downstream tasks while using <10% of the number of parameters that are used in prior fully-tuned state-of-the art approaches. Our ablation studies confirm the efficacy of our approach over other baselines such as PEFT. Code is available at: `https://samar-khanna.github.io/ExPLoRA/`

## 1. Introduction

Pre-training foundation models (Bommasani et al., 2021) for natural language (Brown et al., 2020; Chowdhery et al.,

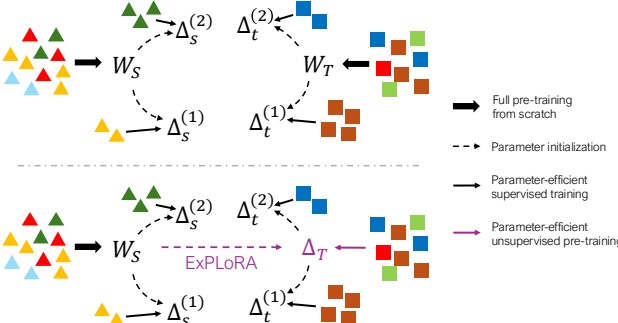

Figure 1: Consider two different image domains, $S$ and $T$. **Above**: the traditional paradigm of pre-training from scratch on each domain $S, T$ to yield $W_S, W_T$, and then fine-tuning on target datasets $i$ to yield $\Delta_s^{(i)}, \Delta_t^{(i)}$. **Below**: our approach, which is to initialize with pre-trained weights from $S$ and then learn unsupervised weights $\Delta_T$ for $T$ in a parameter-efficient manner.

2023; Touvron et al., 2023) and natural images (He et al., 2022; Rombach et al., 2022; Oquab et al., 2024) has historically been computationally intensive, often limited to organizations with substantial resources. However, recent advancements in parameter-efficient fine-tuning (PEFT) techniques including low-rank adaptation (LoRA) and others (Hu et al., 2022; Jia et al., 2022; Qiu et al., 2023) have sparked significant interest. These methods aim to adapt foundation models to downstream supervised-learning tasks using a small fraction (0.1%-10%) of the model's trainable weights, with many based on the hypothesis that the required weight updates to the pre-trained model have a "low intrinsic rank" (Li et al., 2018; Aghajanyan et al., 2020).

In this paper, we focus on visual foundation models (VFMs) such as MAE or DinoV2 (He et al., 2022; Oquab et al., 2024) that were trained on a large amount of natural images. Despite the big investments in developing VFMs for natural images, they underperform when applied to other domains with visual data (e.g. medical or satellite images). For example, fine-tuning a model pre-trained on natural images on satellite image classification tasks is not as effective as fine-tuning one that was pre-trained on satellite images (Ayush et al., 2021; Cong et al., 2022). To bridge this gap, prevailing approaches invest similarly large levels of compute to pre-train VFMs on new domains, inspired by techniques developed for natural images (Cong et al., 2022; Zhou et al.,

[1]Stanford University [2]CZ Biohub. Correspondence to: Samar Khanna <samarkhanna@cs.stanford.edu>.

*Proceedings of the 42nd International Conference on Machine Learning*, Vancouver, Canada. PMLR 267, 2025. Copyright 2025 by the author(s).

2023; Moutakanni et al., 2024; Khanna et al., 2024).

In this work, we challenge this paradigm (fig. 1), asking whether pre-training from scratch on each new domain is strictly necessary, since doing so is expensive (in compute and time) and precludes knowledge transfer from natural images. Instead, we wish to more efficiently leverage the rich semantic information encoded in natural-image VFMs to adapt them to new domains. Our proposed solution addresses these concerns using parameter-efficient self-supervised learning methods for transfer learning.

We introduce **ExPLoRA**, which generalizes vision foundation models to new domains by extending the pre-training phase with parameter-efficient techniques. We initialize a vision transformer (ViT) (Dosovitskiy et al., 2021) with weights pre-trained from natural-image datasets such as MAE or DinoV2. Selectively unfreezing 1-2 transformer blocks, we tune remaining weights with LoRA and continue unsupervised pre-training on the new domain. Subsequently fine-tuning with linear probing or LoRA on this new domain for supervised learning outperforms prior state-of-the-art (SoTA) approaches while training under 5-10% of the original weights. On satellite imagery, we demonstrate an 8% improvement in linear probing top-1 accuracy, and even an improvement over prior SoTA *fully pre-trained and fine-tuned techniques* that required up to 16x more trainable parameters and 8x more compute (GPU hours). We conduct an extensive study on RGB, temporal, and multi-spectral satellite images, either matching or outperforming prior methods that fully pre-train from scratch. ExPLoRA also outperforms previous work on different domains such as wildlife, medical, and agricultural imagery on the WILDS (Koh et al., 2021) benchmark. Our contributions include:

1. Introducing ExPLoRA, a novel parameter-efficient method that extends unsupervised pre-training on target domains, achieving state-of-the-art supervised-learning performance using a fraction of the original ViT weights (section 5).

2. Conducting a comprehensive case study on satellite imagery, outperforming existing techniques on datasets like fMoW and showcasing improvements in linear probing accuracy. We also demonstrate generalization to multiple other domains within WILDS (section 6).

3. Demonstrating ExPLoRA's efficacy via ablation studies and by analyzing improvements in local (positional) and global (class) information encoded in the patch representations output by each ViT block (section 7).

## 2. Related Work

**VFMs**  VFMs such as DinoV2 or masked autoencoders (MAE) that pre-train with self-supervised learning (SSL) have demonstrated remarkable performance across down-stream tasks such as classification or semantic segmentation (Grill et al., 2020; Chen et al., 2020; He et al., 2022; Oquab et al., 2024). However, there has also been a rise in domain-specific VFMs (Cong et al., 2022; Man et al., 2023; Zhang et al., 2023b; Ma et al., 2024; Moutakanni et al., 2024). For instance, SatMAE handles temporal or multi-spectral satellite image inputs. Since these models contain hundreds of millions of parameters, efficient adaptation to downstream tasks has become a key research focus.

**PEFT**  PEFT methods have gained widespread adoption for efficiently adapting large models by updating only a fraction of parameters, mitigating the prohibitive costs of full model tuning. LoRA learns low-rank weight updates to frozen weights, while other methods modify the frequency or number of trainable parameters per layer (Hu et al., 2022; Zhang et al., 2023c; Chavan et al., 2023; Pu et al., 2023). Others use multiplicative orthogonal updates (Qiu et al., 2023; Liu et al., 2024) or inject adapter modules (Chen et al., 2022; Lian et al., 2022; Yin et al., 2023; 2024; Steitz & Roth, 2024), effectively retaining pre-training knowledge in frozen weights. Visual prompt tuning (VPT) methods concatenate learnable prompt tokens to image patch sequences, trading improved fine-tuning performance with increased inference costs (Jia et al., 2022; Yoo et al., 2023; Han et al., 2023; Nie et al., 2023; Tsai et al., 2023; Pei et al., 2024). ExPLoRA aims to supplement rather than replace these methods, and thus can be configured with any existing or future PEFT method for ViT fine-tuning.

**Domain Adaptation**  Domain adaptation enables models trained on a source domain to perform well on a different but related target domain. Traditional transformer-based methods address this via domain alignment, discriminative feature learning, cross-attention with pseudo-labels (Sun et al., 2022; Chuan-Xian et al., 2022; Zhu et al., 2023), or adversarial learning with self-refinement (Xu et al., 2021; Yang et al., 2023), typically requiring labeled target data or source domain labels. Test-time adaptation methods (Wang et al., 2021; Gao et al., 2022; Zhang et al., 2023a) adapt models without target domain labels, but assume a shared label space between domains and thus require models trained with supervised learning on the source domain. While Reed et al. (2022) showed that sequential self-supervised pre-training on source and target datasets improves convergence for supervised tasks, their study was limited to ResNet-50 with MoCo and required full model training. Recent work adapts ViTs through different means: e.g., continual pre-training via masked image modeling (Mendieta et al., 2023) and scaled LoRA adapters (Scheibenreif et al., 2024) for satellite imagery. ExPLoRA builds on this direction, enabling parameter-efficient SSL directly on the target domain.

Further comparisons with related work are in appendix A.

## 3. Background

**MAE** The masked-autoencoder (MAE) (He et al., 2022) is a SSL technique that uses an asymmetric encoder-decoder architecture on images $\mathbf{x} \in \mathbb{R}^{C \times H \times W}$, where patches are masked before being processed by the ViT encoder $f_\theta$, with parameters $\theta$ and layers $\mathcal{L}$. The masked patches are then reconstructed by a smaller decoder $g_\psi$ with weights $\psi$ and layers $\mathcal{L}_D$. $\psi, \theta$ are jointly trained using mean-squared error on the reconstructed visible pixels $(g_\psi \circ f_\theta)(\mathbf{x})$. While effective across domains (Cong et al., 2022; Bachmann et al., 2022), MAEs typically require full fine-tuning for downstream tasks, making them computationally expensive.

**DinoV2** DinoV2 (Oquab et al., 2024) is a robust SSL method for ViTs. Unlike MAE, DinoV2 features $f_\theta(\mathbf{x})$ have demonstrated strong zero-shot performance, enabling adaptation to downstream tasks even with a frozen ViT backbone. During pre-training, DinoV2 maintains two copies of a ViT encoder: the student $f_\theta$ (trainable) and the teacher $f_{\theta'}$, where $\theta'$ is updated using an exponential-moving average of the student's parameters $\theta$. The training objective incorporates a global, image-level loss from Dino (Caron et al., 2021), a patch-based loss from iBOT (Zhou et al., 2021), and regularizers including KoLeo (Delattre & Fournier, 2017) and Sinkhorn-Knopp centering (Caron et al., 2020).

**LoRA** Low-rank adaptation (LoRA) (Hu et al., 2022) assumes that the weight update to change a set of unsupervised pre-trained weights to supervised fine-tuned weights lives in a low-rank subspace,

$$W \approx W_0 + \Delta_W = W_0 + BA \tag{1}$$

where $W \in \mathbb{R}^{k_2 \times k_1}$ are the final, task-specific fine-tuned weights, $W_0 \in \mathbb{R}^{k_2 \times k_1}$ are the pre-trained weights, $\Delta_W \in \mathbb{R}^{k_2 \times k_1}$ is the weight update required to translate the pre-trained weights $W_0$ to the fine-tuned weights $W$. The key is that $\Delta_W = BA$, where $B \in \mathbb{R}^{k_2 \times r}$ and $A \in \mathbb{R}^{r \times k_1}$ form a low-rank factorization of $\Delta_W$, with $r \ll \min(k_1, k_2)$.

## 4. Problem Setup

Consider a set of image domains $\mathcal{D} = \{1, 2, \dots\}$, where each domain $d \in \mathcal{D}$ is associated with a data distribution $p_d(\mathbf{x})$, and images $\mathbf{x} \in \mathbb{R}^{C_d \times H_d \times W_d}$ have domain-specific channel, height, and width. Let $S \in \mathcal{D}$ represent a source domain (e.g., internet-scale natural image data) and $T \in \mathcal{D}$ represent a target domain (e.g., satellite imagery). The data from the source domain follow a distribution $p_S(\mathbf{x})$, and the target domain data come from $p_T(\mathbf{x})$. For the target domain, the joint distributions $p_T^{(\tau)}(\mathbf{x}, \mathbf{y})$ describe images $\mathbf{x}$ with associated supervised labels $\mathbf{y}$ used for each downstream task $\tau$. We then assume access to the following:

(i) $W_S$, pre-trained weights obtained via unsupervised pre-training on images from $p_S(\mathbf{x})$

(ii) $\mathcal{X}_T = \{\mathbf{x}_i\}_{i=1}^N \sim p_T(\mathbf{x})$, an unlabeled dataset of $N$ images from new domain $T$

(iii) $\mathcal{Y}_T^{(\tau)} = \{\mathbf{x}_j, \mathbf{y}_j\}_{j=1}^{M^{(\tau)}} \sim p_T^{(\tau)}(\mathbf{x}, \mathbf{y})$, labeled datasets of $M^{(\tau)}$ images for each task $\tau$ in domain $T$

Our objective is to learn optimal weights $W_T^{(\tau)}$ for each supervised-learning dataset $\mathcal{Y}_T^{(\tau)}$ in a parameter-efficient manner while leveraging the knowledge stored in $W_S$.

Traditionally, the approach (fig. 1) has been to begin pre-training from scratch on data $\mathcal{X}_T$ from the new domain of interest, and then fine-tune for each dataset $\mathcal{Y}_T^{(\tau)}$, given as:

$$W_T^{(\tau)} \approx W_T + \Delta^{(\tau)} \tag{2}$$

where $W_T$ represents the weights learned from unsupervised pre-training on $\mathcal{X}_T$, and $\Delta^{(\tau)}$ are the weights learned from supervised fine-tuning on $\mathcal{Y}_T^{(\tau)}$. However, this method is computationally expensive: fully pre-training $W_T$ from scratch for every new target domain requires prohibitively large amounts of additional compute.

LoRA, however, addresses this inefficiency as follows:

$$W_T^{(\tau)} \approx W_S + \Delta^{(\tau)} = W_S + B^{(\tau)} A^{(\tau)} \tag{3}$$

The LoRA hypothesis is that the update $\Delta^{(\tau)}$ resides in a low-rank subspace when adapting pre-trained weights $W_S$ to fine-tuned weights $W_T^{(\tau)}$. This hypothesis holds well when pre-training and fine-tuning distributions are similar. However, when there is significant domain shift, such as between natural images and multi-spectral satellite data, the low-rank assumption often breaks down (see section 6.1.3).

Our goal is to learn $W_T^{(\tau)}$ efficiently while using the knowledge encoded in $W_S$ to bridge the large domain shift to $T$. We propose the following partition of $W_T^{(\tau)}$:

$$W_T^{(\tau)} \approx W_S + \Delta_T + \Delta^{(\tau)} \tag{4}$$

where $\Delta_T \in \mathbb{R}^{k_2 \times k_1}$ is an additional structured update matrix learned from unsupervised pre-training on $\mathcal{X}_T$. Crucially, $\Delta_T$ requires only a fraction of the $k_1 k_2$ parameters of $W_S$, making it significantly more efficient than full-rank pre-training. The resulting model, $W_T^* = W_S + \Delta_T \approx W_T$, retains the benefits of unsupervised pre-trained VFMs, including strong feature extraction, effective linear probing, KNN classification, and generalization to downstream tasks.

## 5. Method

To learn $\Delta_T$, we propose ExPLoRA (i.e. **Ex**tended **P**re-training with **LoRA**), a method that efficiently adapts a pre-trained ViT to a new target domain $T$, in algorithm 1.

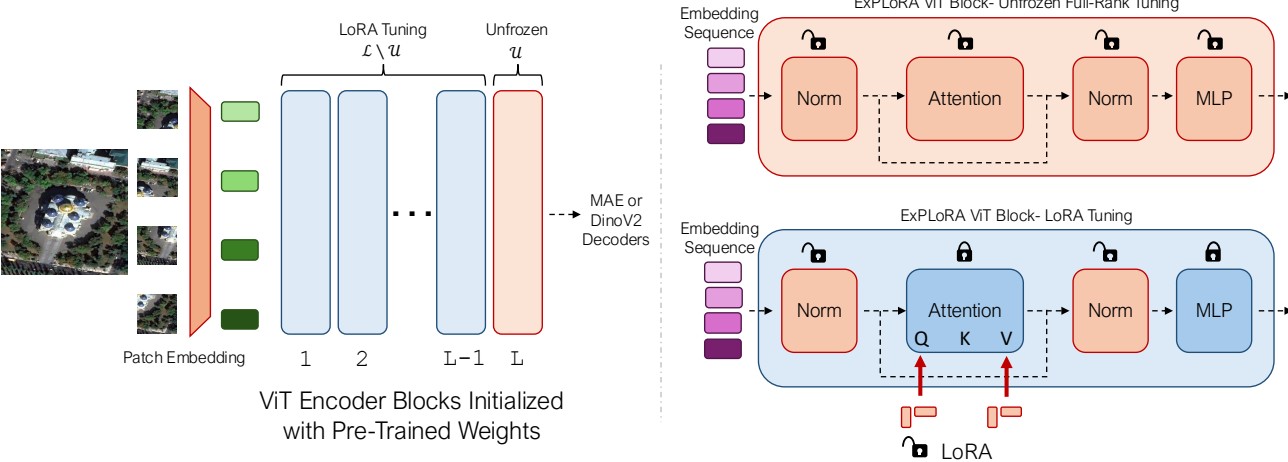

Figure 2: An overview of ExPLoRA. The set $\mathcal{L}$ of L ViT blocks is partitioned into two sets: $\mathcal{U}$ (red), which denotes blocks whose parameters are completely unfrozen, and $\mathcal{L} \setminus \mathcal{U}$ (blue) which denotes blocks that undergo LoRA tuning (only on the $Q, V$ attention matrices). Note that the normalization layers are always unfrozen across all blocks.

---

**Algorithm 1** ExPLoRA

1: **Input:** $W_S \coloneqq$ pre-trained ViT with L layers $\mathcal{L} = \{1, \dots, \text{L}\}$; $\mathcal{X}_T \coloneqq$ unlabeled dataset
2: Initialize a frozen ViT with $W_S$ from source domain $S$ (e.g., DinoV2 or MAE weights).
3: Unfreeze all parameters of a subset of blocks $\mathcal{U} \subset \mathcal{L}$. (e.g., $\mathcal{U} = \{\text{L}\}$ or $\mathcal{U} = \{1, \text{L}\}$).
4: Apply LoRA (with rank $r$) on $Q$ and $V$ weights in attention layers of frozen blocks in $\mathcal{L} \setminus \mathcal{U}$ and unfreeze normalization layers in these blocks.
5: Train all unfrozen parameters $\Delta_T$ on the unlabeled dataset $\mathcal{X}_T$ using the same unsupervised objective as what was used for $W_S$ (e.g., DinoV2 or MAE).
6: **Output:** A new pre-trained model $W_T^* = W_S + \Delta_T$ for target domain $T$.

---

Concretely, we learn $\Delta_T$ as follows:

$$\Delta_T = \underset{\theta \in \Theta(\mathcal{U}, r)}{\arg\min} \left( \min_\psi \sum_{\mathbf{x} \in \mathcal{X}_T} \mathcal{C}_S \left( g_\psi \left( f_\theta(\mathbf{x}; W_S) \right), \mathbf{x} \right) \right) \tag{5}$$

where $f_\theta(\cdot; W_S)$ is a ViT feature encoder parameterized by $\theta$ with initialization $W_S$, $g_\psi$ is a learnable decoder parameterized by $\psi$ (e.g., MAE/DinoV2 decoder), $\mathcal{C}_S$ is the unsupervised loss that was used for $W_S$ (e.g., reconstruction loss), and $\Theta(\mathcal{U}, r)$ restricts the trainable parameter space to full-rank updates for blocks in $\mathcal{U}$ and LoRA rank-$r$ updates in $\mathcal{L} \setminus \mathcal{U}$. Similar to Goyal et al. (2023), we find that using the same unsupervised loss $\mathcal{C}_S$ as was used for $W_S$ is beneficial. As we show in section 6, ExPLoRA can even outperform full pre-training on new domains from scratch.

In terms of notation, D-[L]-$r64$ refers to a ViT initialized with DinoV2 weights, where $\mathcal{U} = \{\text{L}\}$, and LoRA rank 64 is applied to the $Q, V$ matrices of attention layers in $\mathcal{L} \setminus \mathcal{U}$.

Thus, $\Delta_T$ comprises of all weights in $\mathcal{U}$, LoRA matrices in $\mathcal{L} \setminus \mathcal{U}$, and layer normalization parameters of all blocks.

For **DinoV2**, we initialize a ViT-L with $W_S$ from DinoV2's ViT encoder, without registers (Darcet et al., 2024). Since DinoV2 pre-trained checkpoints don't include Dino or iBOT linear heads, we initialize a shared head $g_\psi$ from scratch, which is fully trained during extended pre-training.

For **MAE**, we initialize a ViT-L with $W_S$ from the MAE ViT-L encoder and use provided weights to initialize decoder $g_\psi$ (He et al., 2022). During extended pre-training, beyond the ExPLoRA recipe in algorithm 1, we apply LoRA with rank $r'$ on the $Q, V$ matrices of each attention layer in $g_\psi$. The LoRA rank $r'$ may differ from rank $r$ used in the ViT encoder $f_\theta$ (appendix B.4). All other decoder weights except layer-normalization are frozen, with no block fully unfrozen in $g_\psi$ to minimize trainable parameters.

For the multi-spectral ViT introduced by SatMAE we need to additionally unfreeze the positional encoding and patch embedding weights for each group of channels, as part of $\Theta(\mathcal{U}, r)$ in eq. (5). These cannot be initialized from $W_S$, as $W_S$ is trained on RGB inputs, whereas multi-spectral inputs can have more or different channels.

**Fine-Tuning post-ExPLoRA** After running ExPLoRA, we receive a new unsupervised model $W_T^* = W_S + \Delta_T$ for the target domain $T$ that functions as any other pre-trained ViT $W_T$. $g_\psi$ (e.g., the Dino/MAE decoder) is discarded as it is not part of the ViT encoder $f$. Only $\Delta_T$, consisting of 1-2 unfrozen ViT blocks, LoRA matrices, and layer-normalization weights, are stored for each $T$– all of which can be merged into the original ViT, thus preserving architecture. Like LoRA, ExPLoRA significantly reduces additional storage requirements compared to fully pre-training $W_T$.

Finally, solving for $\Delta^{(\tau)}$ on each labeled dataset $\mathcal{Y}_T^{(\tau)}$:

$$\Delta^{(\tau)} = \underset{\substack{\phi \\ \theta \in \Theta(r)}}{\arg\min} \sum_{(\mathbf{x},\mathbf{y}) \in \mathcal{Y}_T^{(\tau)}} \mathcal{C}' \left( h_\phi^{(\tau)} \left( f_\theta(\mathbf{x}; W_T^*) \right), \mathbf{y} \right) \quad (6)$$

where the ViT $f_\theta$ is now initialized with $W_T^*$, $h_\phi^{(\tau)}(\cdot)$ is a task-specific ViT decoding head with parameters $\phi$, and $\mathcal{C}'$ represents a supervised-learning loss function.

For any PEFT method, we allow $\theta \in \Theta(r)$, where $r$ restricts the trainable parameter space $\Theta$ of $\mathcal{L}$ (e.g., LoRA only on attention $Q, V$ matrices). For linear probing, $\Theta(0) = \emptyset$, so $(h_\phi^{(\tau)} \circ f_\theta)(\mathbf{x}; W_T^*) = \phi^\top f(\mathbf{x}; W_T^*)$. For general fine-tuning, we optimize both $\theta \in \Theta$ and $\phi$ unrestricted, modifying $h^{(\tau)}$ as per the task (e.g., a decoder head for segmentation). With $\Delta^{(\tau)}$, eq. (4) gives us our final model weights $W_T^{(\tau)}$, which can be used for classification, segmentation, detection etc.

## 6. Experiments

Our experimental results consist of a case study on satellite imagery (section 6.1), with an ablation study in section 6.1.2. We evaluate on multiple downstream tasks in sections 6.1.3, 6.1.4 and 6.2. Additional experiments and ablations are provided in appendix B and training hyperparameter and compute configurations are mentioned in appendix C.

We report both performance metrics and computational requirements for our experiments. As a pre-training technique, ExPLoRA offers an efficient alternative to full domain-specific pre-training. Thus, unlike task-specific fine-tuning, its compute costs are amortized across all downstream applications of the resulting model including feature extraction, linear probing, and various supervised tasks.

Our results achieve a new SoTA top 1 accuracy of 79.3% (↑1.5%) on the competitive fMoW-RGB benchmark, outperforming fully pre-trained and fine-tuned models while using 6% of the ViT encoder parameters and requiring only 100 GPU hours compared to 960+ hours for full pre-training. We also achieve a ↑8.2% improvement in linear probing accuracy on the same dataset. Across other satellite datasets, ExPLoRA matches or exceeds fully-pretrained prior state-of-the-art methods while requiring **8x-10x** less compute and **16x** fewer trainable parameters and demonstrates competitive performance on WiLDS benchmark datasets as well.

### 6.1. Case Study: Satellite Imagery

We examine satellite images given their importance towards societal applications (section 8) and since they represent a significant domain shift from natural images. There is a large and growing body of research on developing foundation models for satellite imagery from scratch (Cong et al., 2022; Reed et al., 2023; Tang et al., 2024), thus presenting a good benchmark for ExPLoRA.

### 6.1.1. RGB SATELLITE IMAGES

**Dataset** We first consider the functional map of the world (fMoW) dataset of high-resolution satellite images, each paired with one of 62 classification labels (Christie et al., 2018). fMoW is used as a benchmark for satellite-image foundation models (Cong et al., 2022; Reed et al., 2023).

We compare our results in table 1 against both prior fully pre-trained SoTA foundation models as well as PEFT techniques applied on ViTs pre-trained with MAE and/or DinoV2 weights. Our results demonstrate that D-ExPLoRA-[L]-$r64$ is SoTA in terms of fMoW-RGB average accuracy at **79.28%**. ExPLoRA outperforms techniques that require fully and/or continually pre-training ViTs on fMoW while using 6% of the original ViT encoder parameters and 8x less compute. Further experiments with MAE are in B.6.

ExPLoRA-initializations with LoRA fine-tuning outperform other unsupervised initializations paired with PEFT techniques by 1-3%, including SoTA matrix-adaptation methods like AdaLoRA (Zhang et al., 2023c), BOFT (Liu et al., 2024), VPT approaches such as GVPT (Yoo et al., 2023), SA$^2$VP (Pei et al., 2024), and adapter methods like Adapter+ (Steitz & Roth, 2024). We also outperform satellite image continual pre-training methods such as GFM (Mendieta et al., 2023) and GDA (Scheibenreif et al., 2024) by 6%. Additionally, applying SA$^2$VP to ExPLoRA-initialized ViTs improves performance over DinoV2 by 1% (rows 16 vs 17), showcasing ExPLoRA's compatibility with other PEFT methods and its versatility as an initialization for new domains.

Using our strongest performing variant (i.e. ExPLoRA with DinoV2), we investigate linear-probing performance on fMoW-RGB compared with prior SoTA methods in table 2. Linear-probing represents freezing the backbone and then training a linear head on the features extracted from the frozen backbone, serving as a desirable metric of the quality of extracted embeddings. Our results demonstrate an improvement of over ↑8.2% in top 1 average accuracy over prior SoTA methods, demonstrating that ExPLoRA learns robust unsupervised representations for its target domain without requiring expensive from-scratch pre-training. Importantly, ExPLoRA outperforms domain-specific prior SoTA solutions (rows 1-4), as well as DinoV2, which suggests successful transfer learning on the target domain by leveraging knowledge from pre-training on natural images.

### 6.1.2. ABLATION STUDY

We perform an ablation study (table 3) on linear-probing performance for fMoW-RGB to determine whether our proposed configuration performs optimally.

A natural question is whether the improvement in performance stems primarily from unfreezing blocks, or from

| | Model | PEFT | Pre-Train Params | Fine-Tune Params | Pre-Train GPU hours | Fine-Tune GPU hours | Top 1 Acc. |
|---|---|---|---|---|---|---|---|
| 1 | ScaleMAE [56] | Full | 303.3M | 303.3M | 960 | 610 | 77.80 |
| 2 | SatMAE [17] | Full | 303.3M | 303.3M | 960 | 610 | 77.78 |
| 3 | SatMAE [17] | LoRA-$r8$ [30] | 303.3M | 0.8M | 960 | 220 | 76.10 |
| 4 | ScaleMAE [56] | LoRA-$r8$ [30] | 303.3M | 0.8M | 960 | 220 | 78.01 |
| 5 | GFM [45] | LoRA-$r8$ [30] | 303.3M | 0.8M | 900 | 220 | 73.03 |
| 6 | GDA [58] | GDA-$r16$ [58] | 8.5M | 8.5M | 310 | 310 | 71.88 |
| 7 | MAE [28] | LoRA-$r8$ [30] | – | 0.8M | – | 220 | 76.21 |
| 8 | M-[L]-$r64$ | LoRA-$r8$ [30] | 18.7M | 0.8M | 80 | 220 | 76.55 |
| 9 | DinoV2 [48] | LoRA-$r8$ [30] | – | 0.8M | – | 220 | 78.08 |
| 10 | DinoV2 [48] | BOFT-$b2m8$ [41] | – | 0.9M | – | 230 | 72.40 |
| 11 | DinoV2 [48] | Mona [73] | – | 7.1M | – | 220 | 72.80 |
| 12 | DinoV2 [48] | VPT-100 [31] | – | 0.4M | – | 500 | 77.29 |
| 13 | DinoV2 [48] | GVPT-100 [74] | – | 0.4M | – | 500 | 76.22 |
| 14 | DinoV2 [48] | AdaLoRA-$r8$ [77] | – | 1.2M | – | 220 | 78.87 |
| 15 | DinoV2 [48] | Adapter+ [61] | – | 1.4M | – | 220 | 78.16 |
| 16 | DinoV2 [48] | SA$^2$VP [50] | – | 1.1M | – | 410 | 77.53 |
| 17 | D-[L]-$r64$ | SA$^2$VP [50] | 18.7M | 1.1M | 100 | 410 | 78.51 |
| 18 | D-[L]-$r64$ | LoRA-$r8$ [30] | 18.7M | 0.8M | 100 | 220 | **79.28** |

Table 1: Results on the fMoW-RGB validation dataset. "Pre-train / Fine-tune Params" refer to trainable parameters of the ViT-L encoder required on the *new* domain, i.e. satellite images. M-[L]-$r64$ and D-[L]-$r64$ refer to ExPLoRA models initialized with MAE and DinoV2 weights, respectively (section 5). Our measurements for GPU hours use standardized hardware platforms for fair comparison.

| | Method | Arch. | Top 1 Acc. |
|---|---|---|---|
| 1 | GASSL [2] | ResNet | 68.32 |
| 2 | SatMAE [17] | ViT-L | 65.94 |
| 3 | ScaleMAE [56] | ViT-B | 67.30 |
| 4 | CrossScaleMAE [63] | ViT-B | 69.20 |
| 5 | DinoV2 [48] | ViT-L | 67.60 |
| 6 | DinoV2† [48] | ViT-L | 69.00 |
| 7 | D-[L]-$r64$ | ViT-L | **76.86** |
| 8 | D-[L]-$r64$† | ViT-L | **77.48** |

Table 2: Linear-probing on fMoW-RGB. The first four rows fully pre-train on the dataset. † denotes concatenating features from the last 4 ViT blocks. All other rows use features of the last ViT block.

LoRA-tuning the rest of the ViT. We investigate this by unfreezing blocks {L,L-1} (with no LoRA) in row 4, and comparing that with ExPLoRA-L-$r8$ in row 13. As seen, unfreezing an extra block consumes almost double the number of parameters, but fails to yield the same improvement in performance $\downarrow 0.34\%$. Thus, simply increasing the number of unfrozen blocks will likely improve performance, but will not do so as effectively as ExPLoRA, and will also significantly and sharply decrease the parameter-efficiency.

Next, we investigate whether applying high-rank LoRA to all matrices (including MLP) outperforms targeting only attention $Q, V$ matrices. Surprisingly, LoRA-$r128$ on all matrices (row 6) or only on MLP matrices (row 7) significantly harms performance compared to LoRA-$r256$ on just Q,V matrices (row 5). However, both rows 5 and 6 are much less parameter-efficient than ExPLoRA (rows 13-15).

The choice of $\mathcal{U}$ matters as well. As seen in rows 8-10, and 15, for the DinoV2 objective, $\mathcal{U} = \{1\}$ or $\mathcal{U} = \{9\}$ are not

| | Blocks Unfrozen | LoRA Rank | Norm Unfrozen | LoRA Layers | Num. Params | GPU hours | Top 1 Acc. |
|---|---|---|---|---|---|---|---|
| 1 | – | – | – | – | – | – | 69.00 |
| 2 | All | N/A | Yes | [] | 303.3M | 1200 | 54.29 |
| 3 | [L] | 0 | ✓ | [] | 12.7M | 90 | 74.83 |
| 4 | [L-1,L] | 0 | ✓ | [] | 25.3M | 130 | 75.97 |
| 5 | [] | 256 | ✓ | [Q,V] | 25.9M | 180 | 75.51 |
| 6 | [] | 128 | ✓ | All | 33.1M | 220 | 55.03 |
| 7 | [L] | 64 | ✓ | Mlp | 16.5M | 140 | 48.55 |
| 8 | [1] | 64 | ✓ | [Q,V] | 18.7M | 100 | 75.97 |
| 9 | [9] | 64 | ✓ | [Q,V] | 18.7M | 100 | 75.45 |
| 10 | [L-1] | 64 | ✓ | [Q,V] | 18.7M | 100 | 77.40 |
| 11 | [L] | 0 | ✓ | VPT-100 | 12.8M | 430 | 70.14 |
| 12 | [L] | 64 | ✗ | [Q,V] | 18.6M | 100 | 76.78 |
| 13 | [L] | 8 | ✓ | [Q,V] | 13.4M | 90 | 76.31 |
| 14 | [L] | 32 | ✓ | [Q,V] | 15.7M | 100 | 76.40 |
| 15 | [L] | 64 | ✓ | [Q,V] | 18.7M | 100 | 77.48 |
| 16 | [L-1,L] | 64 | ✓ | [Q,V] | 31.1M | 140 | 77.76 |
| 17 | [1,L-1,L] | 64 | ✓ | [Q,V] | 43.4M | 180 | **78.04** |

Table 3: Ablation study using DinoV2-ExPLoRA, measuring linear-probing accuracy on fMoW-RGB. The second row performs full pre-training from scratch. All results are obtained by using concatenated features from the last 4 ViT blocks.

as effective as $\mathcal{U} = \{L-1\}$ or $\mathcal{U} = \{L\}$, ceteris paribus. To understand this result further, see section 7. We also notice a slight drop in accuracy from leaving the normalization layers across the ViT frozen, seen in row 12.

Lastly, we investigate the impact of LoRA rank and increasing the set of unfrozen blocks $\mathcal{U}$ on ExPLoRA, effectively varying the parameter budget $\Theta(\mathcal{U}, r)$. As seen in table 3, changing the rank from 8 to 32 yields a modest improvement ($\uparrow 0.09\%$, rows 13 vs 14), but increasing

from 32 to 64 brings about a much larger gain ($\uparrow 1.08\%$, rows 14 vs 15), with an equivalent 3M parameter increase. Meanwhile, expanding the set of unfrozen blocks from $\mathcal{U} = \{\texttt{L}\}$ to $\mathcal{U} = \{\texttt{L-1, L}\}$ (rows 15 vs 16) and further to $\mathcal{U} = \{\texttt{1, L-1, L}\}$ (rows 16 vs 17) yield consistent improvements of $\uparrow 0.28\%$.

These results indicate that strategically constraining the parameter space via $\Theta(\mathcal{U}, r)$ in ExPLoRA yields better performance more efficiently than either uniform high-rank updates across all layers (row 5) or simply unfreezing more blocks without LoRA (row 4). This balance between full-rank updates in key transformer blocks and low-rank updates elsewhere more effectively captures domain-specific knowledge during pre-training.

Notably, full pre-training from scratch (row 2) achieves only 54.29% accuracy despite using 1200 GPU hours, suggesting that significantly more compute might be needed for a performant from-scratch DinoV2 checkpoint. In contrast, our standard configuration for ExPLoRA in row 15 achieves 77.48% accuracy with just 100 GPU hours (a $12\times$ reduction in compute) while using only 18.7M parameters (6% of the full model). While rows 16-17 show further performance improvements with additional unfrozen blocks, row 15 represents our preferred trade-off between performance and efficiency for most applications. Further ablations on compute efficiency (B.2), data efficiency (B.3), MAE decoder rank (B.4), and ViT backbone size (B.5) are in appendix B.

### 6.1.3. MULTI-SPECTRAL SATELLITE IMAGES

**Dataset**  Next, we consider the fMoW-Sentinel dataset, a large dataset of Sentinel-2 images introduced by Cong et al. (2022). Each image consists of 13 spectral bands and is paired with one of 62 classes.

With fMoW-Sentinel, we evaluate transfer from natural images to multi-spectral, low-resolution satellite images- a domain with significant distribution shift from RGB images due to the absence of non-RGB bands in $S$. We use the group-channel ViT-L from Cong et al. (2022), initialized with MAE. During ExPLoRA, we additionally unfreeze only the patch embedding layers due to architectural differences.

Table 4 highlights the challenge: direct fine-tuning of MAE weights on this domain results in a substantial 9% performance gap compared to SatMAE (rows 1 vs 2). LoRA tuning from MAE performs worse (row 3), and unfreezing four transformer blocks (row 7) fails to help. ExPLoRA bridges this gap effectively while requiring just 320 GPU hours and <10% trainable parameters for pre-training compared to 1150 hours for SatMAE. This demonstrates ExPLoRA's ability to efficiently adapt to domains with substantial distribution shifts from natural images, preserving performance while dramatically reducing computational requirements.

| | Method | PEFT | PT / FT Params | PT / FT GPU hours | Top 1 Acc. |
|---|---|---|---|---|---|
| 1 | MAE [28] | Full | – / 303.3M | – / 770 | 51.61 |
| 2 | SatMAE [17] | Full | 303.3M / 303.3M | 1150 / 770 | **61.48** |
| 3 | MAE [28] | LoRA-$r8$ | – / 0.8M | – / 290 | 46.97 |
| 4 | SatMAE [17] | LoRA-$r8$ | 303.3M / 0.8M | 1150 / 290 | 59.48 |
| 5 | GFM [45] | LoRA-$r8$ | 303.3M / 0.8M | 960 / 290 | 57.55 |
| 6 | GDA [58] | GDA-$r16$ | 7.3M / 7.3M | 560 / 410 | 55.23 |
| 7 | MAE$\ast$ | LoRA-$r8$ | 51.5M / 0.8M | 380 / 290 | 54.12 |
| 8 | M-$\texttt{[L]}$-$r32$ | LoRA-$r8$ | 16.2M / 0.8M | 290 / 290 | 51.84 |
| 9 | M-$\texttt{[1,L]}$-$r32$ | LoRA-$r8$ | 29.7M / 0.8M | 320 / 290 | **60.15** |

Table 4: Results on fMoW-Sentinel (validation), with ViT-L. "PT" and "FT" refer to pre-training and fine-tuning, respectively. "Params" refers to trainable parameters required on the *new* domain, i.e. multi-spectral satellite images. "MAE$\ast$" refers to initializing a SatMAE model with MAE weights and then pre-training with blocks 1,2,23,24 unfrozen. ExPLoRA achieves competitive performance while requiring significantly less compute than full domain-specific pre-training.

| | Method | PEFT | PT / FT Params | PT / FT GPU hours | Top 1 Acc. |
|---|---|---|---|---|---|
| 1 | GASSL [2] | Full | 23.5M / 23.5M | 380 / 220 | 74.11 |
| 2 | SatMAE [17] | Full | 303.3M / 303.3M | 1120 / 610 | **79.69** |
| 3 | MAE [28] | LoRA-$r8$ | – / 0.8M | – / 250 | 69.30 |
| 4 | SatMAE [17] | LoRA-$r8$ | 303.3M / 0.8M | 1120 / 250 | 75.27 |
| 5 | M-$\texttt{[L]}$-$r32$ | LoRA-$r8$ | 18.7M / 0.8M | 100 / 250 | **75.98** |

Table 5: Classification results on the validation set of fMoW-Temporal. "PT" and "FT" refer to pre-training and fine-tuning, respectively. All MAE/SatMAE experiments use ViT-L, while GASSL uses a ResNet. ExPLoRA (M-$\texttt{[L]}$-$r32$) achieves the best PEFT performance while using only a fraction of the compute required for full pre-training.

### 6.1.4. ADDITIONAL SATELLITE DATASETS

We perform extensive experiments on downstream satellite datasets, with further results in B.1.

**fMoW-Temporal**  Each input is a sequence of up to 3 fMoW-RGB (Christie et al., 2018) images of a location, distributed temporally, and paired with one of 62 classes. Since the inputs are now temporal sequences, we initialize the temporal MAE architecture from Cong et al. (2022) with MAE weights, and pre-train on $\mathcal{X}_T$ with $\mathcal{U} = [\texttt{L}]$ and LoRA rank 32. ExPLoRA then outperforms temporal SatMAE for PEFT (table 5), demonstrating successful transfer learning at a fraction of the pre-training parameters and compute.

**SpaceNet-v1**  This dataset contains high resolution satellite images, each paired with a segmentation mask for buildings (Van Etten et al., 2018). The training and test sets consist of 5000 and 1940 images, respectively. For ExPLoRA, we pre-train on the training set. However, many images in the dataset contain extensive blacked-out regions,

| | Method | PEFT | SpaceNet mIoU | Resisc45 Acc. |
|---|---|---|---|---|
| 1 | SatMAE [17] | Full | 78.07 | 94.80 |
| 2 | ScaleMAE [56] | Full | **78.90** | 95.70 |
| 3 | DinoV2 [48] | LoRA-$r8$ | 76.69 | 97.60 |
| 4 | D-[L]-$r64$ | LoRA-$r8$ | 76.69 | **97.65** |
| 5 | SatMAE [17] | Lin. Probe | 50.89 | 88.30 |
| 6 | ScaleMAE [56] | Lin. Probe | 47.17 | 89.60 |
| 7 | DinoV2 [48] | Lin. Probe | 76.21 | 96.34 |
| 8 | D-[L]-$r64$ | Lin. Probe | **76.34** | **97.32** |

Table 6: Results on the validation sets of SpaceNet-v1, a segmentation task, and Resisc-45, a classification task. The D-[L]-$r64$ ExPLoRA model from the final row of Table 1 achieves state-of-the-art performance on Resisc-45 and competitive results on SpaceNet-v1 without requiring pre-training.

indicating limits of the visible region. Considering this limitation and the small dataset size, it is not clear whether additional pre-training is effective. We find that, despite this, ExPLoRA remains on par with the LoRA-tuned DinoV2 model and remains competitive with the fully pre-trained and fully fine-tuned domain-specific models (table 6).

**RESISC-45** The RESISC-45 (Cheng et al., 2017) benchmark dataset consists of 31,500 satellite images of varying resolution (0.2m-30m GSD), with 45 classes. The data is split into 25,200 training and 6,300 validation images, as per Reed et al. (2023). In table 6, our D-ExPLoRA pre-trained on only high-resolution fMoW-RGB images (last row of table 1) achieves SoTA results of 97.32% on multi-resolution RESISC-45 images, with just linear-probing. We demonstrate successful transfer learning from ExPLoRA pre-training, without requiring any additional modifications for scale-aware representation learning (Reed et al., 2023).

### 6.2. WiLDS Datasets

We test ExPLoRA on the WILDS (Koh et al., 2021) benchmark, specifically on Camelyon17 (Bandi et al., 2018), iWildcam (Beery et al., 2020) and GlobalWheat (David et al., 2020; 2021) datasets, representing domain transfers to medical, wildlife, and agricultural imagery, respectively.

**Camelyon17** The WILDS Camelyon17 dataset consists of images of cancerous and non-cancerous cell tissue. We use the "train-unlabeled" split for pre-training ExPLoRA, and either use LoRA fine-tuning or linear probing on the training set of the labeled split. We report accuracy on the binary classification problem and compare with entries on the WILDS leaderboard which use unlabeled data. Our results in table 7 demonstrate improved performance over domain-specific methods as well as DinoV2, once again successfully bridging the domain gap.

| | Method | PEFT | Pre-Train/Fine-tune GPU hours | Top 1 Acc. |
|---|---|---|---|---|
| 1 | CLater [54] | Full | – | 93.90 |
| 2 | ICON | Full | – | 90.10 |
| 3 | MAE [28] | LoRA-$r8$ | – / 120 | 92.13 |
| 4 | M-[L]-$r32$ | LoRA-$r8$ | 90 / 120 | 92.24 |
| 5 | DinoV2 [48] | Lin. Probe | – / 10 | 93.27 |
| 6 | DinoV2 [48] | LoRA-$r8$ | – / 120 | 92.97 |
| 7 | D-[L]-$r32$ | Lin. Probe | 90 / 10 | **94.41** |
| 8 | D-[L]-$r32$ | LoRA-$r8$ | 90 / 120 | 94.21 |

Table 7: Classification results on the validation set of Camelyon17. ExPLoRA models achieve the best performance in both linear probing and LoRA-based fine-tuning configurations.

| | Method | PEFT | Pre-Train/Fine-tune GPU hours | Top 1 Acc. |
|---|---|---|---|---|
| 1 | MAE [28] | LoRA-$r8$ | – / 90 | 60.07 |
| 2 | M-[L]-$r32$ | LoRA-$r8$ | 70 / 90 | 61.86 |
| 3 | DinoV2 [48] | Lin. Probe | – / 10 | 66.04 |
| 4 | DinoV2 [48] | LoRA-$r8$ | – / 90 | 67.10 |
| 5 | D-[L]-$r32$ | Lin. Probe | 70 / 10 | 62.95 |
| 6 | D-[L]-$r32$ | LoRA-$r8$ | 70 / 90 | **68.07** |

Table 8: Classification results on the validation set of iWildcam.

| | Method | Pre-Train/Fine-tune GPU hours | Top 1 Acc. | AP@ 0.5:0.95 | AR@ 0.5:0.95 |
|---|---|---|---|---|---|
| 1 | ICON [35] | – | 68.9 | – | – |
| 2 | MAE [28] | – / 190 | 82.5 | 53.8 | 58.7 |
| 3 | M-[L]-$r64$ | 30 / 190 | 79.3 | 51.6 | 56.6 |
| 4 | DinoV2 [48] | – / 190 | 82.3 | 52.1 | 57.1 |
| 5 | D-[L]-$r64$ | 30 / 190 | **82.7** | **54.5** | **59.2** |

Table 9: Object detection results on the validation set of Global-Wheat. AP and AR stand for average precision and average recall. ExPLoRA outperforms the baseline methods across all metrics.

**iWildcam** iWildcam classification requires identifying one of 182 animal species given an image. We pre-train on the training set, finding that this outperforms pre-training on the extra-unlabeled set. In table 8, we find an improvement over DinoV2 using LoRA-$r8$ PEFT. Surprisingly, the linear probing performance of the ExPLoRA suffers in comparison with DinoV2, suggesting possible overfitting due to a small domain gap. This is likely because natural image datasets such as ImageNet (Deng et al., 2009) used for pre-training DinoV2 already contain many images of animals.

**GlobalWheat** The GlobalWheat dataset consists of an object detection task, where images of wheat fields are associated with bounding boxes on the visible wheat heads (David et al., 2020; 2021). ExPLoRA extends pre-training on the training set, and then we fine-tune using Detectron2 code for object-detection with ViTs (Wu et al., 2019). ExPLoRA outperforms both fully pre-trained baselines from the WILDS leaderboard and strong VFMs DinoV2 and MAE on top 1 accuracy, average precision, and average recall.

# 7. Further Analysis

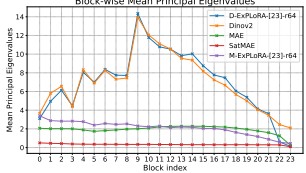 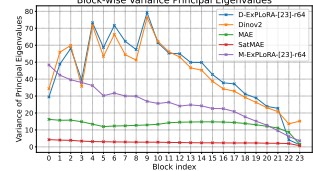

Figure 3: The mean of the eigenvalues of the feature map outputted by each ViT block.

Figure 4: The variance of the eigenvalues of the feature map outputted by each ViT block.

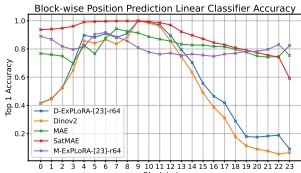 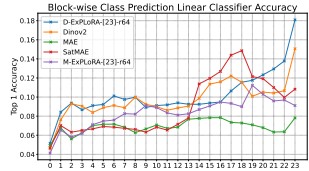

Figure 5: Linear probing patches for position (local information), across all ViT blocks.

Figure 6: Linear probing patches for classification (global information), across all ViT blocks.

The key design choice of ExPLoRA is to constrain the trainable parameter space during pre-training via $\Theta(\mathcal{U}, r)$ in eq. (5), where $\mathcal{U} \subset \mathcal{L}$ layers of the ViT undergo full-rank training while the remaining frozen layers $\mathcal{L} \setminus \mathcal{U}$ receive low-rank updates. For parameter-efficiency, we wish to keep $|\mathcal{U}| \ll |\mathcal{L}|$ and make an informed choice of which layers to unfreeze based on their potential to improve representation learning during extended pre-training.

We conduct an investigation on 5 models using a sample of $\mathcal{X}_{D_T}$. These models are DinoV2, D-ExPLoRA-[L]-$r64$, SatMAE, MAE, and M-ExPLoRA-[L]-$r64$. We do the following analyses: (i) PCA to measure the mean and variance of eigenvalues of patch feature vectors for each ViT block, in figs. 3 and 4 (ii) linear probing for local or global information (Darcet et al., 2024) by training logistic regression classifiers on each block's patch feature vectors, to predict either patch position (fig. 5) or image class (fig. 6).

**Findings and Unfreezing Strategy for DinoV2**: Our analysis reveals that the spectral properties of a block's feature map (fig. 3) and the ability to retrieve local information from its output patch tokens (fig. 5) are correlated. The classification accuracy for position and the mean and variance of the principal eigenvalues peak in the middle-layers of the model, suggesting that the middle blocks capture fine-grained local properties of patches (e.g., texture, relative position). Meanwhile, deeper blocks focus on global semantic understanding, as shown by increased classification accuracy for image class prediction in fig. 6 and lower variance in feature map eigenvalues in fig. 4. Combined, these results suggest that unfreezing deeper layers, such as $\mathcal{U} = \{L\}$, allows the model to better capture global features without overfitting to local details of images of $T$. This is

empirically confirmed in table 3, where linear probing accuracy correlates inversely with the mean eigenvalue of each block (i.e., block 23 > block 22 > block 0 > block 9). The attention maps in fig. 9 further support this, showing that deeper layers focus more clearly on central objects, while earlier layers (e.g., blocks 9, 10) exhibit diffuse attention patterns spread around the border.

**Findings and Unfreezing Strategy for MAE**: For MAE, we see a similar, but less pronounced trend. However, MAE is only trained for reconstruction, and so retains more local information across the ViT's layers. This is reflected by its lower patch-wise eigenvalues, higher localization accuracy, and lower global accuracies than Dino.

**ExPLoRA's Impact**: D-ExPLoRA preserves local information in the middle layers but also improves localization accuracy in the last few layers. Importantly, it also enhances the global information contained in the patches for deeper model layers. This indicates a better understanding of the target domain, as seen in B.8, where ExPLoRA's attention highlights the central object more clearly.

Thus, our analysis provides guidelines for practitioners to select which blocks to unfreeze based on the eigenvalue properties and classification accuracy patterns of different ViT layers, offering a systematic approach to constraining $\Theta(\mathcal{U}, r)$ when pre-training with ExPLoRA (eq. (5)).

# 8. Conclusion and Discussion

In this paper, we introduce ExPLoRA, a novel pre-training strategy to adapt pre-trained ViT foundation models for natural images to additional visual domains such as satellite imagery or medical data. We challenge the common paradigm of expensive pre-training from scratch for each new visual domain by offering a solution to transfer knowledge from foundation models that matches or outperforms domain-specific foundation models. ExPLoRA makes powerful foundation models accessible to researchers with limited computational resources while using 8-10x less compute and 16x fewer parameters. Our hope is that ExPLoRA enables further use of VFMs on domains other than natural images without requiring vast computational resources for pre-training.

While effective, many aspects of ExPLoRA deserve further study. The strategy of fully training a small budget of weights combines well with PEFT techniques like LoRA–understanding this further would be valuable. Future work might explore whether other parameter-efficient techniques could improve ExPLoRA during pre-training more effectively than unfreezing blocks. Lastly, an investigation of ExPLoRA on large language models would be valuable.

## Impact Statement

As the scale of models and datasets grows exponentially, access to the computing power necessary to develop and use foundation models is increasingly restricted to the hands of a few organizations. This leaves many researchers in academia or smaller companies reliant on the resources of such organizations for ML research and applications. Techniques such as PEFT can alleviate this dependence and enable those with fewer computational resources to adapt, investigate, and customize models for their own needs. We hope that ExPLoRA furthers this goal, allowing ML practitioners to tailor foundation models with minimal compute, thus broadening access to powerful ML tools for critical fields like sustainability and medicine.

For example, automated analysis of satellite imagery can inform social, economic, and environmental policies, but manual curation is expensive, and pre-training models on such data has significant costs, both environmental and otherwise (appendix D). ExPLoRA offers a more efficient way to create new foundation models for different visual domains via distilling knowledge from existing foundation models trained on natural images. This can sharply reduce costs, aid researchers and policymakers, and enable flexible downstream applications.

## Acknowledgements

This research is based upon work supported in part by the Office of the Director of National Intelligence (ODNI), Intelligence Advanced Research Projects Activity (IARPA), via 2021-2011000004, NSF(#1651565), ARO (W911NF-21-1-0125), ONR (N00014-23-1-2159), CZ Biohub, HAI. The views and conclusions contained herein are those of the authors and should not be interpreted as necessarily representing the official policies, either expressed or implied, of ODNI, IARPA, or the U.S. Government. The U.S. Government is authorized to reproduce and distribute reprints for governmental purposes not-withstanding any copyright annotation therein.

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

# Appendix

We include supplementary material in the following sections. We provide additional contextualization and comparison with recent related work in appendix A, additional experimental results and ablations in appendix B, training and hyperparameter details in appendix C, and an overview of the environmental impact of our work in appendix D.

## A. Further Contextualization with Related Work

Here, we expand upon ExPLoRA's differences with recent related work.

### A.1. Comparison with Geospatial Domain Adaptation

Recent work has explored both continual pre-training (GFM) (Mendieta et al., 2023) and parameter-efficient domain adaptation (GDA) (Scheibenreif et al., 2024) for satellite imagery. We compare these approaches with ExPLoRA in table 10.

ExPLoRA differs from these approaches in several key aspects. Unlike GFM which trains the full backbone, ExPLoRA achieves superior performance with only a fraction of trainable parameters. While GDA is also parameter-efficient, it requires non-mergeable scaling vectors that induce inference latency and modify the ViT, whereas ExPLoRA's LoRA adapters can be merged into the ViT's weights. Additionally, ExPLoRA extends beyond MAE architectures (supporting DinoV2 and others) and allows flexible configurations between pre-training and fine-tuning, including varying LoRA ranks or using different PEFT methods, which GDA doesn't support out-of-the-box.

| | GFM [45] | GDA [58] | ExPLoRA (Ours) |
|---|---|---|---|
| Parameter-efficient | ✗ | ✓ | ✓ |
| Training objective | MAE | MAE | Any |
| Arch. preservation | ✓ | ✗ | ✓ |
| Fine-tuning | Any PEFT | Only LoRA | Any PEFT |

Table 10: Differences between prior geospatial domain adaptation methods and ExPLoRA

We also demonstrate ExPLoRA's broader applicability through experiments on larger datasets (fMoW-RGB, fMoW-Sentinel, which have 400k-800k images vs 90k images in FireRisk (Shen et al., 2023), the largest dataset used in GDA) and domains beyond remote sensing (i.e. WiLDS). Our analysis in section 7 provides insights into block-wise information encoding, offering practitioners a systematic approach for block selection during extended pre-training– a unique feature not present in prior work.

### A.2. Comparison with Unsupervised Domain Adaptation

Unsupervised domain adaptation (UDA) enables models to generalize to unseen domains (Kang et al., 2019; Oren et al., 2019; Singhal et al., 2023; Khanna et al., 2023). Traditional UDA assumes:

(i) $\mathcal{Y}_S = \{\mathbf{x}_i, \mathbf{y}_i\}_{i=1}^{N'} \sim p_S(\mathbf{x}, \mathbf{y})$, a labeled source domain dataset

(ii) $\mathcal{X}_T = \{\mathbf{x}_i\}_{i=1}^{N} \sim p_T(\mathbf{x})$, an unlabeled target domain dataset

(iii) $Y_T \subseteq Y_S$, constraining the label-set of $T$ with respect to $S$

| | UDA | ExPLoRA |
|---|---|---|
| Source data | Labeled | None |
| Source knowledge | Data | Weights |
| Target data | Unlabeled | Unlabeled |
| Label constraints | $Y_T \subseteq Y_S$ | None |

Table 11: Differences between UDA and ExPLoRA

Common UDA benchmarks like Office-Home (Venkateswara et al., 2017) and VisDA-2017 (Peng et al., 2017) follow this setup (Xu et al., 2021; Sun et al., 2022; Yang et al., 2023; Zhu et al., 2023).

ExPLoRA's setting in section 4 is **different**: we only require *weights* $W_S$ from unsupervised pre-training on $p_S(\mathbf{x})$, without source data access or label set restrictions. This enables adaptation across wider domain shifts (e.g., ImageNet to multi-spectral satellite imagery, section 6.1.3). Thus, rather than competing with UDA methods, ExPLoRA can complement them by providing a better initialization than standard natural-image pre-training (as seen empirically in appendix B.7).

## B. Additional Experimental Results

Below, we include further experimental results as a continuation of section 6.

### B.1. Results on Additional Downstream Datasets

**NAIP**  We consider a land-cover classification dataset used in (Ayush et al., 2021), where each of 244,471 training and 55,529 validation images are paired with one of 66 land cover classes obtained by the USDA's National Agricultural Imagery Program. In table 12, we first demonstrate similar performance between both natural-image backbones (rows 4 and 5), which surprisingly outperform SatMAE, which is pre-trained on fMoW-RGB. We use ExPLoRA to pre-train from DinoV2 to the training set of this dataset (without labels). Our results (row 6) demonstrate comparable performance, suggesting that for this dataset, domain-specific knowledge may not be highly relevant to successfully solve the task.

| | Method | PEFT | Top 1 Acc. |
|---|---|---|---|
| 1 | GASSL [2] | Full | 57.63 |
| 2 | SatMAE [17] | Full | **71.77** |
| 3 | SatMAE [17] | LoRA-r8 | 69.45 |
| 4 | MAE [28] | LoRA-r8 | 70.36 |
| 5 | DinoV2 [48] | LoRA-r8 | **70.40** |
| 6 | D-`[L]`-$r32$ | LoRA-r8 | **70.40** |

Table 12: NAIP validation set results

**EuroSAT**  The dataset contains 27,000 13-band satellite images of 10 classes (Helber et al., 2019), sourced from Sentinel-2. For ExPLoRA, we don't pre-train on this dataset's training set, and instead use LoRA fine-tuning starting with the pre-trained weights learned in the last row of table 4. We demonstrate improved performance over DinoV2, and match the performance achieved by the domain-specific SatMAE which was fully pre-trained on fMoW-Sentinel, and fully fine-tuned on EuroSAT (table 13). This demonstrates the successful use of our extended pre-trained model on further downstream datasets.

| | Method | PEFT | Top 1 Acc. |
|---|---|---|---|
| 1 | SeCo [44] | Full | 93.14 |
| 2 | SatMAE [17] | Full | 98.98 |
| 3 | SatMAE [17] | LoRA-r8 | **98.73** |
| 4 | DinoV2 [48] | BOFT-b8m2 | 96.60 |
| 5 | M-`[1,L]`-$r64$ | LoRA-r8 | 98.54 |

Table 13: EuroSAT validation set results

### B.2. The Importance of Extended Pre-training

To evaluate ExPLoRA's effectiveness, we analyze how its performance scales with computational resources. Specifically, we investigate two key questions: first, given a fixed compute budget, what is the optimal allocation between extended pre-training and fine-tuning? Second, for a fixed parameter budget, does investing compute in extended pre-training provide advantages over standard fine-tuning approaches?

We address these questions in fig. 7, focusing on DinoV2 models running on NVIDIA-A4000 GPUs. We evaluate D-ExPLoRA-[L]-$r64$ for different lengths of pre-training (50k, 100k, 150k, and 200k iterations), corresponding to 24, 48, 72, and 96 GPU-hours of extended pre-training respectively. Each checkpoint undergoes LoRA-$r8$ fine-tuning. We compare against three baselines: (i) Direct LoRA-$r8$ fine-tuning on DinoV2 weights (ii) Fine-tuning DinoV2 with block 24 unfrozen and LoRA-$r64$ (matching ExPLoRA's parameter budget) (iii) Fine-tuning DinoV2 with blocks 0, 1, 23, 24 unfrozen and LoRA-$r64$ (55.8M parameters vs ExPLoRA's 18.7M).

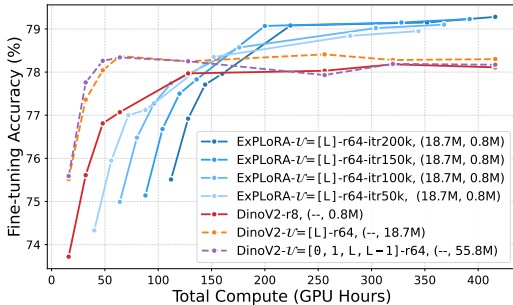

Figure 7: Fine-tuning accuracy versus total compute (measured in GPU-hours). Total compute includes both pre-training (if applicable) and fine-tuning phases. Along with the label for each method in the legend, we include (#pre-training params, #fine-tuning params).

Results in fig. 7 demonstrate that ExPLoRA's extended pre-training achieves a ↑ 1.0% improvement in maximum fine-tuning accuracy within the same total compute budget (320 GPU hours). Notably, even increasing the parameter budget during fine-tuning fails to match this performance. While additional pre-training iterations beyond 100k improve initial fine-tuning accuracy, they have minimal impact on the final accuracy ceiling, highlighting ExPLoRA's computational efficiency. Importantly, the total amount of compute invested in pre-training and fine-tuning ExPLoRA is still dwarfed by the compute spent for full pre-training from scratch, e.g., 960 GPU hours for SatMAE (table 1).

Lastly, we re-iterate that the benefits of extended pre-training become more pronounced as the domain gap increases. While fig. 7 demonstrates a 1% improvement on RGB satellite images, table 4 highlights a 8% improvement when using ExPLoRA + PEFT (row 8) over full fine-tuning MAE (row 1) on multi-spectral satellite images– a domain with substantial shift from natural images. ExPLoRA still requires less total compute (610 vs 770 GPU hours), demonstrating that its efficiency advantages scale with domain difficulty.

### B.3. Convergence and Data Efficiency

Another important question is on ExPLoRA's data efficiency- i.e. can ExPLoRA achieve good representations on the target domain without requiring many training iterations?

In fig. 8, we plot the linear-probing accuracy against the number of extended pre-training iterations for ExPLoRA (in blue). ExPLoRA improves quickly, requiring between 100-150k extended pre-training iterations to reach optimal performance. As discussed in section 6.1.2, unfreezing additional transformer blocks (in red) fails to achieve the same level of performance while requiring more parameters.

One hypothesis for the effectiveness of pairing unfreezing blocks with LoRA is that low-rank updates to the ViT backbone "nudge" the sequence of embedded visual tokens from $S$ to those representing $T$, which then enables the unfrozen ViT block to efficiently compress global information from the new domain.

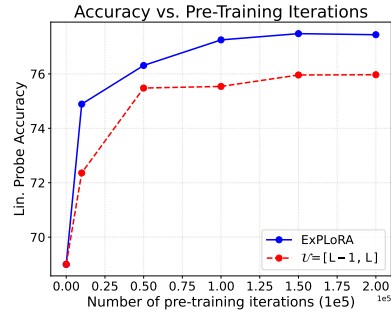

Figure 8: Lin. probe accuracy vs. number of training iterations.

### B.4. Impact of MAE Decoder Rank

As outlined in section 5, we initialize the MAE decoder $g_\psi$ (with its $\mathcal{L}_D$ transformer layers) with pre-trained weights $W_S$ from He et al. (2022), keeping all decoder weights (except layer norm) frozen during extended pre-training on $\mathcal{X}_T$. We apply LoRA with rank $r'$ to the $Q, V$ weights of the attention layers in the decoder $\mathcal{L}_D$, while unfreezing 1-2 blocks $\mathcal{U}$ in the ViT encoder $\mathcal{L}$ and applying LoRA with rank $r$ to the remaining layers $\mathcal{L} \setminus \mathcal{U}$ (algorithm 1).

We evaluate ExPLoRA with M-[1, L]-$r64$ on fMoW-Sentinel, using a fixed encoder LoRA rank $r = 64$, unfreezing blocks $\mathcal{U} = \{1, L\}$, and varying the decoder rank $r'$. We then fine-tune the resulting model with LoRA $r = 8$ and measure the highest top 1 accuracy on the validation set of fMoW-Sentinel. Table 14 shows that increasing $r'$ up to 32 improves fine-tuning performance, which then declines by $\downarrow 0.94\%$ for $r' = 64$. This suggests that balancing the unfrozen parameters between the ViT encoder $f_\theta$ (used for fine-tuning) and the MAE decoder $g_\psi$ (discarded post pre-training) is crucial. Larger $r'$ may improve the decoder's ability without benefiting the learned representations of $f$. This issue doesn't arise in DinoV2, as the Dino-iBOT shared head is fully trained since it isn't provided by Oquab et al. (2024).

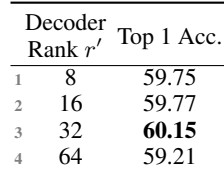

|   | Decoder Rank $r'$ | Top 1 Acc. |
|---|---|---|
| 1 | 8 | 59.75 |
| 2 | 16 | 59.77 |
| 3 | 32 | **60.15** |
| 4 | 64 | 59.21 |

Table 14: Ablation on M-[1, L]-$r64$ on the validation set of fMoW-Sentinel. Here, the LoRA rank used for the ViT-L encoder is fixed at $r = 64$, while the rank $r'$ for MAE decoder is varied.

### B.5. Impact of ViT backbone size

We also test the impact of the ViT backbone for ExPLoRA, varying the architecture for DinoV2 from ViT-B (86M, L = 12 layers, embedding dimension 768), ViT-L (303M parameters, L = 24 layers, embedding dimension 1024), and ViT-G (1100M parameters, L = 40 layers, embedding dimension 1280) for extended pre-training on fMoW-RGB. The ExPLoRA models we compare against are D-[12]-$r64$ for ViT-B, D-[24]-$r64$ for ViT-L, and D-[32]-$r32$ for ViT-G. We unfreeze the 12th, 24th, and 32nd layers for each of ViT-B, ViT-L, and ViT-G, picking these layers by extending the analysis from section 7 to ViT-B and ViT-G. We find that the 12th (last layer) for ViT-B and the 32nd (out of 40) layer for ViT-G output representations with low mean eigenvalues compared to other layers, thus presenting good candidates for unfreezing.

|   | Method | Arch. | Top 1 Acc. Last 1/Last 4 |
|---|---|---|---|
| 1 | DinoV2 [48] | ViT-B | 63.62/65.90 |
| 2 | DinoV2 [48] | ViT-L | 67.60/69.00 |
| 3 | DinoV2 [48] | ViT-G | 70.07/70.36 |
| 4 | D-[12]-$r64$ | ViT-B | 74.72/75.11 |
| 5 | D-[24]-$r64$ | ViT-L | 76.86/77.48 |
| 6 | D-[32]-$r32$ | ViT-G | **77.29/77.79** |

Table 15: Linear probing results on fMoW-RGB (validation), where we vary the size of the ViT encoder $\mathcal{L}$ from ViT-B, ViT-L, and ViT-G. "Last 1/Last 4" refers to using the output representation from just the last 1 or the last 4 ViT layers.

In table 15, we see that as expected, ViT-G performs the best, but is only $\uparrow 0.31\%$ better in top 1 accuracy compared to ViT-L, while using many more parameters. On the other hand, we see the highest impact for ExPLoRA on ViT-B, where the top 1 accuracy improves by $\uparrow 9.21\%$ over the original DinoV2 ViT-B. These results further demonstrate the effectiveness and efficiency of ExPLoRA as a powerful technique to create unsupervised foundation models for new visual domains.

## B.6. Additional PEFT baselines for MAE

As a continuation of table 1, we include PEFT methods used on MAE weights, which generally underperform compared with DinoV2. For completeness, these results are in table 16.

| | Model | Arch. | PEFT | Pre-Train Params | Fine-Tune Params | Pre-Train GPU hours | Fine-Tune GPU hours | Top 1 Acc. |
|---|---|---|---|---|---|---|---|---|
| 1 | SatMAE [17] | ViT-L | LoRA-$r8$ [30] | 303.3M | 0.8M | 960 | 220 | 76.10 |
| 2 | MAE [28] | ViT-L | LoRA-$r8$ [30] | – | 0.8M | – | 220 | 76.21 |
| 3 | MAE [28] | ViT-L | DVPT-10 [31] | – | 0.4M | – | 500 | 72.35 |
| 4 | MAE [28] | ViT-L | GVPT-100 [74] | – | 0.4M | – | 500 | 70.86 |
| 5 | MAE [28] | ViT-L | SA$^2$VP [50] | – | 1.1M | – | 410 | 73.55 |
| 6 | MAE [28] | ViT-L | AdaLoRA-$r8$ [77] | – | 1.2M | – | 220 | 75.25 |
| 7 | MAE [28] | ViT-L | Adapter+ [61] | – | 1.4M | – | 220 | 74.10 |
| 8 | MAE [28] | ViT-L | Mona [73] | – | 7.1M | – | 220 | 74.76 |
| 9 | M-[L]-$r64$ | ViT-L | LoRA-$r8$ [30] | 18.7M | 0.8M | 80 | 220 | **76.55** |

Table 16: MAE+PEFT results on fMoW-RGB validation split (table 1, contd.). "Pre-train Params" and "Fine-tune Params" refer to trainable parameters of the ViT encoder required on the *new* domain (satellite images).

Pre-training ExPLoRA with MAE is slightly more efficient than with DinoV2 (i.e. 80 vs 100 GPU hours) due to two factors: the MAE encoder only operates on visible image patches, and unlike DinoV2, it does not require maintaining a separate "teacher" model copy.

## B.7. Results on UDA Benchmarks

As discussed in appendix A.2, while ExPLoRA is not a traditional unsupervised domain adaptation (UDA) method, it can serve as an initialization for ViT-based UDA approaches. We demonstrate this compatibility below.

| | Method | Arch. | Init. | plane | bcycl | bus | car | horse | knife | mcycl | person | plant | sktbrd | train | truck | Mean |
|---|---|---|---|---|---|---|---|---|---|---|---|---|---|---|---|---|
| 1 | CDTrans [70] | DEiT | IN | 97.1 | 90.5 | 82.4 | 77.5 | 96.6 | 96.1 | 93.6 | **88.6** | **97.9** | 86.9 | 90.3 | **62.8** | 88.4 |
| 2 | PMTrans [80] | ViT-B | IN-21k | 98.9 | **93.7** | 84.5 | 73.3 | 99.0 | 98.0 | 96.2 | 67.8 | 94.2 | **98.4** | 96.6 | 49.0 | 87.5 |
| 3 | TVT [71] | ViT-B | IN-21k | 97.1 | 92.9 | 85.3 | 66.4 | 97.1 | 97.1 | 89.3 | 75.5 | 95.0 | 94.7 | 94.5 | 55.1 | 86.7 |
| 4 | TVT* [71] | ViT-B | IN-21k | 95.8 | 85.8 | 81.9 | 68.4 | 95.9 | 96.2 | 91.9 | 70.3 | 93.8 | 93.7 | 92.9 | 48.5 | 84.6 |
| 5 | TVT [71] | ViT-B | DinoV2 | 98.9 | 88.7 | 90.3 | 64.2 | **99.3** | 74.5 | 95.3 | 66.0 | 85.3 | 94.6 | **97.9** | 54.6 | 84.1 |
| 6 | TVT [71] | ViT-B | **Ours** | 97.0 | 89.9 | 89.4 | 73.8 | 98.0 | 88.9 | 94.4 | 85.9 | 93.8 | 94.5 | 97.7 | 54.3 | **88.2** |
| 7 | SSRT [62] | ViT-B | IN-21k | 98.9 | 87.6 | 89.1 | 84.8 | 98.3 | 98.7 | 96.3 | 81.1 | 94.9 | 97.9 | 94.5 | 43.1 | 88.8 |
| 8 | SSRT [62] | ViT-B | DinoV2 | 99.2 | 88.1 | 89.9 | 85.4 | 98.4 | 98.9 | **97.6** | 84.8 | 96.2 | 97.1 | 95.4 | 48.3 | 89.9 |
| 9 | SSRT [62] | ViT-B | **Ours** | **99.4** | 88.6 | **91.4** | **87.9** | 98.3 | **99.1** | 97.1 | 88.0 | 95.9 | 98.1 | 96.0 | 51.2 | **90.9** |

Table 17: Classification accuracy (%) on VisDA-2017 (validation). Results marked with * use our reproduced results. Using ExPLoRA initialization improves UDA performance compared to standard ImageNet initialization.

**VisDA2017**   The VisDA2017 dataset (Peng et al., 2017) contains 152,297 training and 55,388 validation images across 12 object classes. The dataset represents a synthetic-to-real domain shift: training images are synthetically rendered 3D models under various lighting conditions, while validation images are sourced from MS-COCO (Lin et al., 2014).

Table 17 shows ExPLoRA's effectiveness when combined with TVT (Yang et al., 2023) and SSRT (Sun et al., 2022), two state-of-the-art UDA methods. Using ExPLoRA D-[12]-$r64$ (DinoV2-initialized ViT-B with last layer unfrozen and LoRA-r64 elsewhere) pre-trained on both synthetic and real domains, we outperform traditional ImageNet-21k (Deng et al., 2009) initialization by 1-3% while achieving more balanced per-class accuracy. Most notably, when using SSRT, we achieve a new SoTA accuracy of **90.9%**, surpassing both ImageNet and DinoV2 initialization by substantial margins (↑2.1% and ↑1.0% respectively). These results are particularly significant as they show ExPLoRA's *unsupervised* initialization can outperform methods that rely on supervised ImageNet-21k pre-training. The benefits of ExPLoRA initialization are also clear with TVT, where performance rises to 88.2% to match recent SoTA methods (Xu et al., 2021; Zhu et al., 2023)– a marked improvement over both its original results (↑2.1%) and DinoV2 initialization (↑4.1%). These results demonstrate that ExPLoRA serves as a powerful initialization strategy for UDA, effectively bridging domain gaps while remaining computationally efficient.

## B.8. Attention Map Visualizations

To aid our analysis in section 7, we visualize attention scores for different ViT blocks across multiple models, including DinoV2, D-[L]-$r64$ (i.e. the last row of table 3), the second and third rows of table 3, MAE, SatMAE, and M-[L]-$r64$. These visualizations are shown in fig. 9 for 3 different images from the validation set of fMoW-RGB. Since our models are trained without registers, we truncate attention scores more than 5 standard deviations away from the mean, thus removing artifact attention scores with unusually high values on background patches (Darcet et al., 2024).

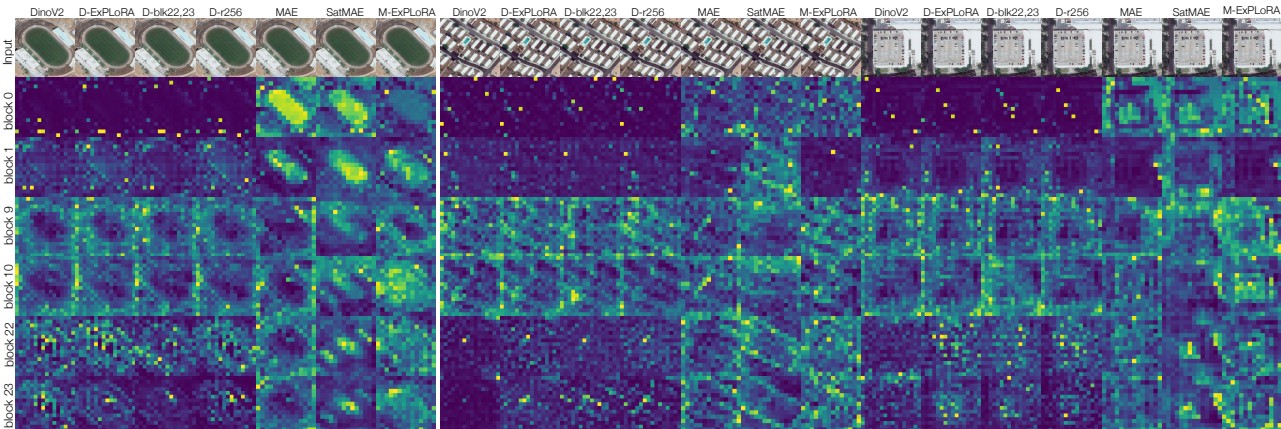

Figure 9: Attention maps visualized from the validation set of fMoW-RGB. The models considered, from left to right, are: DinoV2, D-ExPLoRA-[L]-$r64$, Dino with blocks 22,23 unfrozen during extended pretraining, Dino with LoRA-r256 during extended pre-training, MAE, SatMAE, and M-ExPLoRA-[L]-$r64$. We visualize the attention maps at the beginning, middle, and end blocks of the ViT-L.

The visualizations in fig. 9 further support the analysis in section 7. For the Dino models, the attention scores of block 9-10 are diffuse and spread around the central object of the image, with quite a few border pixels highlighted. Conversely, the attention scores of the final layers are concentrated more towards the central object. These visualizations further suggest that the middle layers focus on capturing local properties of the images such as texture, while the final layers capture global semantic information such as object-ness. Interestingly, the initial blocks for the Dino models display sparse attention patterns with spikes on seemingly random patches. This might suggest a form of caching to aid the computation of deeper layers that will extract local or global information.

For the MAE models, we see that the original MAE (pre-trained on natural images) seem to highlight more border pixels in the final layers of the ViT. Post extended pre-training with ExPLoRA, the final layers concentrate attention scores on the central object, more closely resembling the patterns of SatMAE (which was fully pre-trained on satellite images). ExPLoRA is thus able to successfully transfer knowledge from its initialized source-domain weights $W_S$ to serve as a foundation model $W_T^*$ on the new target domain $T$.

## C. Training Details

In this section, we describe hyperparameters and hardware configurations used for our models. We also include details on datasets used for our experiments.

### C.1. Pre-Training

We use the ViT-Large architecture for all experiments. Since raw image sizes vary, the shorter image size is resized to 224 while preserving aspect ratio, and then a center crop is taken to yield images of size $3 \times 224 \times 224$, representing the channels, height, and width. All pre-training is done on a single NVIDA-RTX 6000 Ada GPU, or 4 NVIDIA-RTX A4000 GPUs on an academic GPU cluster.

**ExPLoRA with MAE** Most of the hyperparameters we use for M-ExPLoRA pre-training follow those in He et al. (2022); Cong et al. (2022). We use an effective batch size of 1024 (through gradient accumulation), a base learning rate of $4.5 \times 10^{-4}$, no weight decay, and a warmup and decaying cosine scheduler, with a warmup of 1 epoch, and a total training time of 200 epochs. We use a masking ratio of $0.75$ and we use the `norm_pix_loss` flag for the MSE loss.

**ExPLoRA with DinoV2**   Most of the hyperparameters for D-ExPLoRA follow the defaults set by Oquab et al. (2024). That is, local (small) crops are between 5%-32% of the original image and are resized to 98x98 pixels, and global (large) crops are greater than 32% of the image and resized to 224x224 pixels. We share the parameters of the Dino-iBOT linear head (3 layers), with a bottleneck dimension of 256, a hidden dimension of 2048, and an output dimension of 65536, initialized from scratch. For Dino, we use Sinkhorn-Knopp (Caron et al., 2020) centering and Koleo (Delattre & Fournier, 2017) regularization with a weight of 0.1. For iBOT, we use masking ratios between 0.1 and 0.5 to mask half of the samples in the batch. The teacher model uses an initial EMA rate of 0.994, with a cosine warmup to 1.000 by the end of training. The teacher warmup and final temperatures are 0.04 and 0.07. The linear Dino-iBOT head is frozen for the first 3k training iterations. We train with the AdamW optimizer (no weight decay), with a base learning rate of $2 \times 10^{-3}$ that is varied with a linear warmup and cosine decay schedule. Training is completed within 200,000 iterations, with a batch size of 32 and with 32 gradient accumulation steps (equalling an effective batch size of 1024), and with an epoch length set to 1000.

### C.2. PEFT Fine-Tuning

We fine-tune using 4 NVIDIA-RTX A4000 GPUs. We use a base learning rate of $10^{-3}$, a cosine scheduler with warmup for 1 epoch, and train for 120 epochs. We use an effective batch size of 256, making use of gradient accumulation if the GPU cannot fit the full batch size in memory.

For data augmentations, we only use the drop-path augmentation (Larsson et al., 2016) at a rate of 0.2, with no dropout, mixup, or cutmix. We note that the original LoRA configuration outperforms other PEFT techniques when paired with the drop-path regularization technique. For example, we find that BOFT does not pair well with drop-path, instead performing most effectively with a custom multiplicative dropout technique (Liu et al., 2024). We include the result with the best hyperparameter configuration for each row in table 1.

### C.3. Linear Probing

We use a single NVIDIA-RTX A4000 GPU for linear probing. We adapt the code provided by Oquab et al. (2024) for linear probing, with a batch size of 256 and a collection of different learning rates: $\left[1 \times 10^{-4}, 1 \times 10^{-3}, 5 \times 10^{-3}, 1 \times 10^{-2}, 2 \times 10^{-2}, 5 \times 10^{-2}, 1 \times 10^{-1}\right]$. We evaluate both probing on average pooled features as well as on the [CLS] token, and also use output features from just the last block, or the last 4 blocks. All numbers reported represent the best validation set accuracy from the best performing configuration.

### C.4. Multi-Spectral Images

We use the group-channel ViT-L architecture introduced in (Cong et al., 2022). We don't use DinoV2 since there is no such architecture for DinoV2 pre-training. Input images are $13 \times 98 \times 98$, representing 13 multi-spectral bands. We follow the configuration in Cong et al. (2022) of dropping bands B1, B9, B10, and use the same grouping strategy. When loading MAE weights to the ViT-L encoder, the patch embeddings do not match and so the patch embedding and group channel encodings are trained from scratch. All other configuration details are the same as for M-ExPLoRA in appendix C.1, except that we use a base learning rate of $4.5 \times 10^{-4}$ for pre-training and train for 50 epochs (given the larger dataset size) on 4 NVIDIA RTX A4000 GPUs for 80 hours.

Fine-tuning details are the same as in C.2.

### C.5. Dataset Information

We include detailed information about the datasets used in our experiments. Table 18 shows the number of samples in the train and validation splits for all datasets used in this work.

Hyperparameter and training configuration details are the same as in appendix C.1 if the images are RGB, and the same as in appendix C.4 if the images have more channels or are temporal.

Since each dataset contains a varying number of training images, the number of ExPLoRA pre-training iterations should be adjusted accordingly. We include the recommended number of ExPLoRA pre-training iterations in table 18. Note, however, that this number can be varied depending on other hyperparameters such as batch size, learning rate, LoRA rank, and number of unfrozen blocks.

| Dataset | #Train | #Validation | ExPLoRA Iters | Domain |
|---------|--------|-------------|---------------|--------|
| fMoW-RGB | 363.6k | 53.0k | 200k | Remote Sensing |
| fMoW-Sentinel | 712.9k | 84.9k | 80k | Remote Sensing (Multi-spectral) |
| fMoW-Temporal | 83.4k | 14.2k | 80k | Remote Sensing (Temporal) |
| SpaceNet V1 | 6.0k | 1.5k | 10k | Remote Sensing |
| Resisc-45 | 18.9k | 6.3k | – | Remote Sensing |
| NAIP | 244.4k | 55.5k | 200k | Remote Sensing |
| EuroSAT | 16.2k | 5.4k | – | Remote Sensing (Multi-spectral) |
| Camelyon17 | 302.4k | 33.6k | 200k | Medical Imaging |
| iWildcam | 129.8k | 7.3k | 150k | Wildlife |
| GlobalWheat | 2.9k | 0.4k | 80k | Agricultural |
| VisDA2017 | 152.3k | 55.4k | 200k | Synthetic-to-Real |

Table 18: Dataset splits and sizes used in our experiments, as well as suggested number of ExPLoRA pre-training iterations.

In general, we find that ExPLoRA demonstrates the largest performance gains on larger datasets where sufficient diversity prevents overfitting during extended pre-training. For datasets with fewer than 50k training samples (e.g., SpaceNet V1, GlobalWheat), the improvements from ExPLoRA are more modest, as the limited data diversity can lead to overfitting with extensive pre-training. Conversely, ExPLoRA works exceptionally well on larger datasets such as fMoW-(RGB, Sentinel, Temporal), Camelyon17, and VisDA2017, where 100k-200k pre-training iterations provide substantial performance improvements while maintaining computational efficiency compared to full domain-specific pre-training.

The licenses for all datasets are included in the footnotes: fMoW[1], Sentinel-2[2], EuroSAT[3], SpaceNet[4], Camelyon17[5], iWildCam[6], GlobalWheat[7].

## D. Environmental Impact

Following (Cong et al., 2022), we compare the carbon footprint of pre-training using ExPLoRA with domain-specific solutions such as SatMAE. We use the carbon footprint calculator proposed by Lacoste et al. (2019). Our results are in table 19[8]

| Method | fMoW-RGB | | fMoW-Sentinel | | fMoW-Temporal | |
|--------|----------|--|---------------|--|---------------|--|
| | GPU hours | kg $CO_2$ eq. | GPU hours | kg $CO_2$ eq | GPU hours | kg $CO_2$ eq. |
| SatMAE | 768 | 109.44 | 576 | 82.08 | 768 | 109.44 |
| ExPLoRA | **96** | **12.44** | **320** | **19.35** | **100** | **12.96** |

Table 19: The estimated carbon footprint of pre-training on these datasets

Since we initialize with pre-trained weights on natural image domains, ExPLoRA is much less environmentally impactful while achieving similar or higher levels of performance. We achieve a 4x-8x reduction in total carbon emitted for each of the large pre-training satellite image datasets considered in table 19.

---

[1]fMoW license: https://github.com/fMoW/dataset/raw/master/LICENSE

[2]Sentinel-2 license: https://scihub.copernicus.eu/twiki/pub/SciHubWebPortal/TermsConditions/Sentinel_Data_Terms_and_Conditions.pdf

[3]EuroSAT license: https://creativecommons.org/licenses/by/4.0/

[4]SpaceNet v1 license: http://creativecommons.org/licenses/by-sa/4.0/

[5]Camelyon17 license:https://creativecommons.org/publicdomain/zero/1.0/

[6]iWildCam license:https://cdla.dev/permissive-1-0/

[7]GlobalWheat license:https://opensource.org/licenses/MIT

[8]Note: while the SatMAE paper reported 768 GPU hours for pre-training on fMoW-RGB and 576 GPU hours on fMoW-Sentinel, our measurements in Tables 1-5 of the main text have a revised number to ensure consistency of hardware platforms across comparisons.

