# OpenReview forum: "ExPLoRA: Parameter-Efficient Extended Pre-Training to Adapt Vision Transformers under Domain Shifts"
_ICML.cc/2025/Conference — ICML 2025 poster_

### Official Review · Reviewer_XYuv · 2025-03-13

**Overall Recommendation:** 3

**Summary:**

The authors show that continued pre-training (with PEFT) on the target domains before supervised fine tuning is an effective methodolofy for adapting natural-image foundation models to non-natural image target tasks. They show particular gains on RGB tasks, but also show some benefit on multi-spectral imagery.

**Claims And Evidence:**

* Claim: Continued pre-training on target domain, with peft, unlocks peft finetuning for specialized tasks.
This is supported in Table 1 and other results tables. However, the question of whether PEFT is needed in either stage is useful (beyond memory savings) is not answered.

**Essential References Not Discussed:**

[1] and [2] are relevant and not discussed.

[1] Goyal, Sachin, et al. "Finetune like you pretrain: Improved finetuning of zero-shot vision models." Proceedings of the IEEE/CVF Conference on Computer Vision and Pattern Recognition. 2023.
[2].    Reed, Colorado J., et al. "Self-supervised pretraining improves self-supervised pretraining." Proceedings of the IEEE/CVF Winter Conference on Applications of Computer Vision. 2022.

**Experimental Designs Or Analyses:**

Reading the paper, the experimental design mostly makes sense, with a few caveats:

Table 2 shows results for both Dinov2 and MAE, but Table 4- 9 don't always show both. Does it still work there?

There is no comparison that I can find for a model which has continued pre-training and finetuning on the same dataset with full finetuning (not ScaleMAE or SatMae), so the performance charachteristics of using PEFT are not well developed.

**Methods And Evaluation Criteria:**

Broadly, the method and evaluations both make sense.

**Other Comments Or Suggestions:**

My main suggestions for improving are discussing the missed references, demonstrating that PEFT is truly needed (not just full finetune), and  sure to show both Dino and MAE results in Table 4-9 for consistency.

**Other Strengths And Weaknesses:**

Strenghts:

Empirically, it works!
Relatively easy to read.

Weaknesses:

In principle, this modernizes "Self-supervised pretraining improves self-supervised pretraining."  with a stronger backbone and peft finetuning, which is not very novel.

**Questions For Authors:**

Here are the points which would lead me to the a higher score.

* Demonstrating that one can't do the proposed two stage finetuning with no PEFT, since PEFT mechnaism is one of the key innovation . I believe this to be true, since it is difficult to not catastrophically forget, but it is the key innovation in the paper, so important to verify

* Show additional evidence in Table 4 ,5 for D -[L}-r32 and M-[L]-r32 in Tables 7-9.

**Relation To Broader Scientific Literature:**

This paper builds on the idea of "finetuning like you pretrain"[1] and "Self-supervised pretraining improves self-supervised pretraining."[2], and adds a PEFT component. They verify this works for adapting natural image models to OOD data domains like sattelite imagery.


[1] Goyal, Sachin, et al. "Finetune like you pretrain: Improved finetuning of zero-shot vision models." Proceedings of the IEEE/CVF Conference on Computer Vision and Pattern Recognition. 2023.
[2].    Reed, Colorado J., et al. "Self-supervised pretraining improves self-supervised pretraining." Proceedings of the IEEE/CVF Winter Conference on Applications of Computer Vision. 2022.

**Theoretical Claims:**

There are no theoretical claims in this paper.

---

> ### Author Rebuttal · Authors · 2025-03-31
>
> Thank you for your feedback and recognition of the empirical performance and novelty of using parameter-efficency for pre-training in ExPLoRA. Here are our responses:
>
> **Q: Is PEFT needed in either pre-training or fine-tuning beyond memory savings?**
> Our primary motivation for parameter-efficient techniques is _efficiency_:
> * Reduced memory footprint provides significant compute savings with larger batch sizes and/or fewer GPUs, and since gradients for <10% of parameters need to be stored and propagated. This makes ExPLoRA much faster than full pre-training (see [revised Table 1](https://imgur.com/a/iMjgOHv) and [response to reviewer W3i1](https://openreview.net/forum?id=OtxLhobhwb&noteId=AapZp7BqgM))
> * Memory savings enable using models such as ViT-L or ViT-G, otherwise inaccessible with constrained GPU budgets
> * While not our main motivation, constraining the parameter space mitigates catastrophic forgetting from the natural image domain. ExPLoRA outperforms fine-tuning from MAE/DinoV2 or domain-specific SatMAE models by >1% in Table 1, and full continual pre-training methods (eg: GFM) by 5-7%. For linear probing, this gap widens to 8%, suggesting benefits from both original and extended pre-training.
>
> We've included a baseline for fully pre-training a ViT-L from scratch on fMoW-RGB with DinoV2 objective in our [revised Table 3](https://imgur.com/a/tWdN4xU). This shows full pre-training cannot match ExPLoRA while requiring >10x compute and >16x parameters.
>
> Lastly, our parameter-efficient pre-training is orthogonal to fine-tuning method choice. ExPLoRA preserves the ViT architecture, enabling any fine-tuning approach. While we primarily use PEFT for efficiency, ExPLoRA also outperforms fully fine-tuned SatMAE and ScaleMAE models by 1.4% (Table 1).
>
> **Q: Are there baselines that fully fine-tune natural-image pre-trained models on the domain of interest?**
> We use PEFT primarily for efficiency. At your request, we've included fully fine-tuning DinoV2 and ExPLoRA models on fMoW-RGB alongside the SatMAE paper's fully fine-tuned MAE result in our [expanded Table 1](https://imgur.com/a/iMjgOHv).
>
> Full fine-tuning is viable with sufficient resources. ExPLoRA models benefit from it just as well as other baselines and outperform them. However, LoRA-r8 PEFT is much cheaper and equally effective. Table 4 includes a comparison where MAE is directly fully fine-tuned on fMoW-Sentinel, showing poor performance (>9% gap) due to the large domain shift.
>
> We reiterate that we don't aim to replace full or PEFT fine-tuning, both complementary to ExPLoRA. Instead, we provide an efficient alternative to _full, domain-specific pre-training_.
>
> **Q: Relation to “self-supervised pre-training improves...” (HPT)**
> We will update our paper to include this reference. Key differences between ExPLoRA and HPT:
>
> * HPT uses two full backbone pre-training phases on different datasets. ExPLoRA uses a single efficient extended pre-training phase that generalizes to multiple downstream tasks.
> * While HPT offers tuning only batch norm layers in the second phase, it centers around full-rank pre-training, like GFM. ExPLoRA uses parameter-efficient techniques that **reduce compute by >10x and parameters by >16x** while achieving similar or higher performance.
> * HPT studies MoCo with ResNet-50. ExPLoRA addresses ViTs with different SSL objectives (DinoV2, MAE), demonstrating broader applicability.
>
> ExPLoRA goes beyond "modernizing HPT" by demonstrating parameter-efficient extended pre-training's value via strong performance at reduced costs across diverse datasets, SSL objectives, and tasks.
>
> **Q: Relation to “Finetune like you pretrain...” (FLYP)**
> We agree this reference is also valuable. As above, there are important differences with ExPLoRA:
> * FLYP is a fine-tuning technique while ExPLoRA is a pre-training technique
> * FLYP focuses on contrastive SSL methods with ResNet. ExPLoRA is compatible with any ViT SSL objective (as we show with DinoV2, MAE).
>
> FLYP does further justify continuing pre-training with the same loss function (eq. 5). We'll update our final manuscript to include this reference.
>
> **Q: Show additional evidence in Table 4 ,5 for D -[L}-r32 and M-[L]-r32 in Tables 7-9.**
> We have now included MAE results in tables 7-9, [linked here](https://imgur.com/a/lx4TaLZ), which underperforms our SoTA Dino-ExPLoRA.
>
> For tables 4-5, we use MAE SSL from SatMAE. There's no DinoV2 modification for multi-spectral/temporal data in prior work. Creating one would require modifying the ViT and Dino SSL mechanism for non-RGB temporal/multi-spectral sequences, which is out of scope. Our goal is to use the SatMAE SSL architecture with MAE weights, showing ExPLoRA matches/outperforms full pre-training (0.67%) and vastly outperforms full MAE fine-tuning on multi-spectral data by 8.54% (Table 2).
>
> ---
> We hope these answers address your questions. If your main concerns are resolved, we kindly request you reconsider your score.

---

> > ### Comment · Reviewer_XYuv · 2025-04-02
> >
> > I am satisfied witht the rebuttal, and therefore raise my score. However, I still view this as very closely related to HPT, and encourage a careful poistioning of your work as it relates to this reference for the camera ready.

---

> > > ### Author Response · Authors · 2025-04-06
> > >
> > > Thank you for your time in reviewing our work and for your helpful suggestions! We are glad to know our rebuttal has resolved your main concerns. For the camera-ready, we will make sure to include a discussion on HPT as related work. We appreciate that you have increased your rating.

---

### Official Review · Reviewer_83s5 · 2025-03-14

**Overall Recommendation:** 2

**Summary:**

This paper proposes a continual pre-training method with a parameter-efficient fine-tuning (PEFT) module such as LoRA (Hu et al. 2021) to improve the adaptability of visual foundation models on specific domains. By inserting and training the PEFT module inside the general-domain pre-trained backbone model with the same learning objective that the backbone model was trained, the proposed ExPLoRA induces meaningful initialization for adaptation with labeled data, resulting in somewhat better classification accuracy on downstream tasks compared with the full fine-tuning approach.







Hu et al. 2021, LoRA: Low-Rank Adaptation of Large Language Models

**Claims And Evidence:**

The authors claim extended pre-training with LoRA (and a few unfreeze model blocks) via self-supervised learning (SSL) objectives can induce better initialization for specific domain datasets. This claim was well-validated through the conventional evaluation protocol of SSL, i.e., evaluation with linear probing and fine-tuning performance on detection and calibration tasks across diverse domains.

**However**, there is a point that can be further improved.
* The author explains the rationale of continual pertaining with LoRA based on weight decomposition, i.e., the desired target weight vector $W_{T}^{(\tau)}$ is the summation of pre-trained weight vector $W_{S}$ on source domain (general knowledge), a vector contains general knowledge on the target domain, $\Delta_{T}$, and the vector represent the task-specific knowledge on target domain $\Delta^{(\tau)}$.
* If the author can provide some qualitative analysis that illustrates the difference (maybe in the embedding space) between pre-trained weight and ExPLoRA-extended pre-trained weight, this design motivation of ExPLoRA will become more convincing.

**Essential References Not Discussed:**

I would recommend the author mention test-time adaptation (TTA) approaches (especially the PEFT-based TTA method) somewhere in the main body of the paper. In terms of the goal -- enabling efficient adaptation of backbone model to target domain -- and design of the learning algorithm, TTA approaches are well-aligned with the position of this work.

- Wang et al. 2020, Tent: Fully Test-time Adaptation by Entropy Minimization
- Gao et al. 2022, VISUAL PROMPT TUNING FOR TEST-TIME DOMAIN ADAPTATION
- Zhang et al. 2023, DomainAdaptor: A Novel Approach to Test-time Adaptation
- Tsai et al. 2023, Convolutional Visual Prompt for Robust Visual Perception

**Experimental Designs Or Analyses:**

Overall the amount and range of experiments are extensive and well-design, but it would be better if the authors could (1) add some qualitative analyses such as embedding visualization, and baseline for the test-time adaptation method

- Wang et al. 2020, Tent: Fully Test-time Adaptation by Entropy Minimization
- Zhang et al. 2023, DomainAdaptor: A Novel Approach to Test-time Adaptation

**Methods And Evaluation Criteria:**

The proposed method is reasonable to address the stated problem -- developing domain adaptable foundation models, and the adopted metric follows the standard of literature.

**Other Comments Or Suggestions:**

See the above reviews.

# Post-rebuttal
> I appreciate the authors' professional rebuttal. For now, I will adhere to my rating because I do not agree with authors' statements on the benefits of the proposed method from the perspective of computational efficiency <-> accuracy improvement trade-off. I will go through other rebuttals as well to reconsider my recommendation (if necessary) through the remaining discussion period.

**Other Strengths And Weaknesses:**

Although the proposed method consistently shows strong results across diverse downstream tasks and domains (`strength`), it requires a non-trivial amount of additional training hours, e.g., 200 hours to achieve marginally improved downstream performance (`weaknesses`).

**Questions For Authors:**

See the above reviews.

**Relation To Broader Scientific Literature:**

The authors claim that the proposed paradigm can indeed contribute to developing the domain-adaptable foundation models by reducing the development cost compared with the from-scratch training approach. However, as shown in Figure 7 of the appendix, the proposed method requires a non-trivial amount of additional training compared with the vanilla pre-trained model for the marginal improvement of downstream performance. Thus the contribution seems not significant.

**Theoretical Claims:**

There are no formal theorems.

---

> ### Author Rebuttal · Authors · 2025-03-31
>
> Thank you for your valuable feedback. We appreciate your recognition of our extensive, well-designed experimental validation and for ExPLoRA’s improved performance over baselines. Here are our responses:
>
> **Q: Can you provide qualitative analysis that illustrates the difference between pre-trained vs ExPLoRA pre-trained embedding spaces?**
> Section 7 contains our analysis of patch embeddings output by ViT blocks across different models (ExPLoRA, DinoV2, MAE, SatMAE), revealing important quantitative differences. Please also see our response to reviewer Fy1r [linked here](https://openreview.net/forum?id=OtxLhobhwb&noteId=JMCCQ90U3c).
>
> Figure 9 (appendix B.8) also qualitatively analyzes attention maps across different ViT blocks of these models. ExPLoRA concentrates attention more tightly on central objects, especially in final layers. This correlates with lower mean eigenvalues and higher positional/classification accuracies from linear probing.
>
> **Q: Include discussion of test-time adaptation (TTA) methods.**
> Thank you for mentioning these methods. We have updated our related work section to include these references which will be present in our camera-ready revision. Our work has key differences with TTA methods:
>
> * TTA methods make a critical assumption: the label space Y is shared between the source and target domains. This makes [1,2,3] incompatible with unsupervised pre-trained backbones when domains have different label sets (common in many settings, including ours). Unsupervised pre-training techniques like DinoV2 or ExPLoRA don't assume specific downstream label sets, requiring some supervised adaptation (linear-probing/PEFT) to parameterize $p^{(\tau)}_T(y|x)$ for downstream task $\tau$ in target domain $T$. TTA methods could then be applied on top of ExPLoRA backbones just as with any ViT. This means that ExPLoRA is complementary with TTA methods rather than a competing method.
> * [3,4] are tailored for CNNs, not applicable to our work with ViTs, which consistently outperform CNNs on our datasets (eg: fMoW-RGB, fMoW-Sentinel, temporal images etc. as verified in prior work).
> * [2] uses visual-prompt tuning (VPT) for ViTs, but lacks a public codebase. We include multiple recent VPT baselines in Table 1 (VPT [5], GVPT [6], SA2VP [7]). ExPLoRA outperforms all by ~2%, including a pre-training baseline with VPT in Table 3.
>
> **Q: ExPLoRA requires a non-trivial amount of additional training compared to fine-tuning a natural-image pre-trained model for an improvement in performance (Figure 7)**
> We would like to refer you to our response to reviewer W3i1, [linked here](https://openreview.net/forum?id=OtxLhobhwb&noteId=AapZp7BqgM).
>
> To summarize, we realize that our plot in figure 7 paints an incomplete picture. ExPLoRA is a pre-training method, providing an alternative to domain-specific _full pre-training_. While extended pre-training + fine-tuning requires more GPU hours than directly fine-tuning, the extended pre-training phase is amortized across multiple downstream tasks, since the same ExPLoRA model can be re-used for initialization. Further, it can be used for methods such as feature extraction, linear probing etc. that a task-specific fine-tuned model cannot do. For linear probing, ExPLoRA outperforms DinoV2/prior SoTA models by >8%. This difference is not captured via figure 7, since that only compares with fine-tuning.
>
> Thus, the fairer comparison is with domain-specific full-pretraining. Here, ExPLoRA requires **8x-10x less compute, 16x fewer parameters** and achieves similar or higher performance. Our expanded [Table 1](https://imgur.com/a/iMjgOHv) demonstrates that ExPLoRA is vastly more efficient than other pre-training methods.
>
> Figure 7 shows only fMoW-RGB performance. On fMoW-Sentinel (larger domain shift to multi-spectral data), full-finetuning a natural-image baseline shows a 9% performance gap with ExPLoRA (row 1, Table 4). Combined, these are significant results since ExPLoRA achieve/surpass fully pre-trained baselines using ~8-10x less compute and 10-16x fewer parameters.
>
> ---
>
> Thank you for your feedback which has strengthened our work. If your concerns are addressed, we kindly ask that you reconsider your score.
>
> ---
> References:
> [1] Tent: Fully Test-time Adaptation by Entropy Minimization, _ICLR 2021_.
> [2] Visual Prompt Tuning for Test-Time Domain Adaptation, _arxiv 2210.04831_.
> [3] DomainAdaptor: A Novel Approach to Test-time Adaptation, _ICCV 2023_.
> [4] Convolutional Visual Prompt for Robust Visual Perception, _NeurIPS 2023_.
> [5] Visual prompt tuning. _ECCV 2022_.
> [6] Improving visual prompt tuning for self-supervised vision transformers. _ICML 2023_.
> [7] SA²VP: Spatially Aligned-and-Adapted Visual Prompt. _AAAI 2024_.

---

> > ### Comment · Reviewer_83s5 · 2025-04-07
> >
> > > Again, I thank the authors for their kind rebuttal, which somewhat addresses my concerns. Although I left a post-rebuttal comment in my review a few days ago, I am commenting on this based on the request from AC for further discussion on efficiency-accuracy trade-off.
> >
> > In summary, **I still don't agree with the advantage of the proposed method in terms of its efficiency**.
> > * The authors claim that ExPLoRA should be compared with other domain-specific pre-training methods, such as ScaleMAE and SatMAE, because ExPLoRA is a kind of pre-training method.
> >   * However, my opinion is that **non-domain-specific methods such as DinoV2 + LoRA (78.08) or DinoV2 + AdaLoRA-r8 (78.87) are already much powerful than domain-specific methods, ScaleMAE/SatMAE (77.80), so why should we compare ExPLoRA with a much inferior method, ScaleMAE?** (the value was parsed from Table 1)
> >   * Maybe the domain-specific baselines the authors considered in this work are too weak, or it is not necessary to pre-train the model entirely on domain-specific data. The non-domain-specific method, DinoV2 + AdaLoRA-r8 (78.87), is already very strong, but the authors' proposed method requires hundreds more GPU hours to achieve a minor improvement (78.87) -> (79.28).
> >     * This raises a question: is the domain-specific extended pertaining really necessary? For example, DinoV2 + AdaLoRA-r8 (78.87) performs better than the authors' domain-specific method `D-[L]-r64 + SA2VP` (78.51) which implies that better architecture and training technique can be much more important than domain-specific extended training.
> >   * **Should the pre-training methods be only compared with other pre-training methods? I think this is not.** The end goal of this domain-specific training is to achieve good performance on downstream tasks from specific domains. In that sense, if the general method (non-domain-specific) already achieves strong performance on the target domains, I think we should regard that general method as a baseline. That's why I mentioned the test-time adaptation (TTA) method as well in my initial review, because TTA also has the same goal as the proposed method, i.e., improvement in performance on specific domains, while the technical approach is different.
> > * I don't mean that ExPLoRA has no contribution to domain-specific model developments. It seems to be an interesting approach worthy to be exploration. However, the practical usefulness of the proposed method is highly questionable given a relatively minor improvement but huge additional computations compared with finetuning-after-natural-image-pretraining baselines. That's why I think this paper's contribution is not strong enough to recommend this paper towards acceptance, and I believe this paper should be polished to improve its practical usefulness.
> >
> > _I am so sorry to say this to authors too late (after AC's action), so that they can not gain any opportunity to refute this. Therefore, although I personally feel a strong weakness in the accuracy-efficiency trade-off, and disagree with the authors' statement for the fair comparison, feel free to downweight my review, AC!_
> >
> > Truly sorry again.
> > Reviewer 83s5

---

> > > ### Author Response · Authors · 2025-04-09
> > >
> > > Thank you for your follow-up. Although we've had very limited time for this response, we appreciate the opportunity to offer clarifications.
> > >
> > > **Q: Why should we compare ExPLoRA with pre-training baselines? Is domain-specific pre-training really necessary?**
> > > ExPLoRA is fundamentally a pre-training method, meant to produce models for multiple downstream tasks without requiring labeled data.
> > >
> > > Our results and multiple prior works cited in our paper clearly demonstrate that domain-specific pre-training is increasingly necessary as the domain gap widens from natural images:
> > >
> > > 1. While DinoV2+AdaLoRA shows competitive performance only on fMoW-RGB (closer to natural images), natural image models significantly underperform
> > >    * On multi-spectral imagery (Table 4): **8-14%** performance gap compared to ExPLoRA and domain-specific models
> > >    * On temporal satellite imagery (Table 5): **6% gap** compared to ExPLoRA
> > >    * On linear probing (Table 2): ExPLoRA shows an **8% improvement**, demonstrating much higher unsupervised embedding quality critical for tasks like clustering/compression. This substantial gap highlights ExPLoRA's ability to capture meaningful domain-specific features without supervised labels.
> > >
> > > 2. Pre-training is functionally different from fine-tuning-- it doesn't require expensive human labels and produces models that can be repurposed for multiple downstream tasks (eg: supervised PEFT, feature extraction), with compute amortized across all applications. This is particularly valuable for domains such as satellite/medical imagery where labeling is more expensive and requires specialized expertise [1]. Please see [this response](https://openreview.net/forum?id=OtxLhobhwb&noteId=AapZp7BqgM) for more.
> > >
> > > 3. ExPLoRA achieves these domain-specific improvements while requiring **>16x fewer parameters and >8x less compute** than fully pre-trained baselines (see [compute augmented Tables](https://imgur.com/a/4xxC3bO)). This efficiency gain is significant for users with limited compute who still need domain-specific models for multiple downstream tasks.
> > >
> > > **Q: DinoV2 + AdaLoRA-r8 (78.87) performs better than D-[L]-r64 + SA2VP (78.51)**
> > > This isn't a valid comparison as it varies both initialization and PEFT method. When keeping the PEFT method constant (SA2VP), ExPLoRA clearly improves over DinoV2 by 1% (78.51% vs. 77.53%). Our best configuration, ExPLoRA+LoRA-r8, achieves 79.28% - outperforming all baselines including DinoV2+AdaLoRA and sets a **new state-of-the-art result** on the competitive fMoW-RGB benchmark.
> > >
> > > What may appear as "minor improvements" (0.4-1%) in aggregate metrics can translate to significant real-world impacts in high-stakes domains like satellite/medical images.
> > >
> > > **Q: Should the pre-training methods be only compared with other pre-training methods?**
> > > No- in fact, our paper includes comprehensive comparisons with both pre-training and fine-tuning approaches. Tables 1, 4-9 all evaluate downstream task performance across various methods.
> > >
> > > However, when assessing computational efficiency, it's appropriate to compare pre-training methods with each other because they serve the same function - producing general-purpose backbones usable for _multiple downstream tasks_ without requiring task-specific labels. This is fair as it reflects how these methods would be used in practice.
> > >
> > > **Q:  TTA also has the same goal as the proposed method, while the technical approach is different**
> > > While both improve specific domain performance, TTA and ExPLoRA have different assumptions:
> > > - TTA assumes shared label spaces between domains
> > > - ExPLoRA produces unsupervised backbones without label space assumptions
> > >
> > > These approaches are complementary, not alternatives.
> > >
> > > **Q: The practical usefulness given relatively minor improvement but huge additional computations**
> > > ExPLoRA's practical utility lies in replacing full domain-specific pre-training while improving performance. The computational demands are modest- ExPLoRA requires only 100 GPU hours for RGB and 300 GPU hours for multi-spectral data, which is:
> > > - Less than the 200-600 fine-tuning GPU hours required in many cases ([see augmented Table 1](https://imgur.com/a/4xxC3bO))
> > > - 10x less compute, 16x fewer parameters, and 8x smaller carbon footprint than full domain-specific pre-training
> > > - A one-time cost that benefits all downstream applications
> > >
> > > The practical benefits include:
> > > - Using ViT-L or ViT-G models on commodity GPUs due to reduced memory/compute footprint
> > > - Allowing researchers with limited resources to customize pre-training for powerful domain-specific models
> > > - Feature extraction with 8% better representations for downstream applications
> > >
> > > For applications in satellite, medical or agricultural monitoring, ExPLoRA's improvements translate to meaningful impact at a fraction of traditional costs (Impact statement).
> > >
> > > ---
> > > We appreciate your engagement, and we hope that our response provides some clarification to your concerns.
> > >
> > > ---
> > > [1] SatMAE, _NeurIPS 2022_.

---

### Official Review · Reviewer_Fy1r · 2025-03-14

**Overall Recommendation:** 3

**Summary:**

The paper proposes ExPLoRA, a parameter-efficient way to extend pre-training of a large vision transformer from its original domain (e.g. natural images) to a new domain (e.g satellite imagery). ExPLoRA accomplishes this by unfreezing 1-2 transformer blocks for full training and applying low-rank (LoRA) updates on the rest. After this unsupervised adaptation, the model can then be fine-tuned using LoRA on labeled tasks and achieves results that match or surpass fully trained domain-specific models.

**Claims And Evidence:**

In general the paper is sound and the claims are supported by enough evidence. However there are two claims to which I would like to draw attention to:
“As seen, un- freezing an extra block consumes almost double the number of parameters, but fails to yield the same improvement in performance ↓ 0.34%. Thus, simply increasing the number of unfrozen blocks will likely improve performance, but will not do so as effectively as ExPLoRA, and will also significantly and sharply decrease the parameter-efficiency”
In section 6.1.2, the authors provide an ablation study where they attempt to unfreeze blocks at different positions (primarily L and/or L-1). While the table supports this claim, it does not provide sufficient evidence to justify choosing only one block to unfreeze. This table is crucial, as their method depends on having at least one block unfrozen to maintain performance. Therefore, more extensive experimentation with various block positions and different numbers of unfrozen blocks would be valuable, as well as replicating these findings across additional datasets.
“Our results demonstrate an improvement of over ↑8.2% in top 1 average accuracy over prior SoTA methods”
In section 6.1.1, the authors make this claim regarding results from satellite images. This is among the paper's strongest claims (also highlighted in the abstract) and is presented in Table 2. However, the comparison appears unfair because it contrasts Dinov2 pretrained weights without domain adaptation against ExPLoRA, which includes domain adaptation. While insightful, this comparison does not clearly illustrate the main use of their method, which aims at efficient in-domain pretraining.
Additionally, I notice the absence of a Dinov2 model fully pretrained in-domain in Table 1. Including such a model would provide a more equitable comparison, as MAE baselines cannot directly be compared with Dinov2.

**Essential References Not Discussed:**

None

**Experimental Designs Or Analyses:**

Experimental designs and analyses are overall adequate. They compare with many different PEFT methods, showing the superiority of their approach in this regard.
See section “Claims And Evidence” for missing experiments/baselines.

**Methods And Evaluation Criteria:**

They perform extensive analysis in a specific use case which is satellite images. Besides, they report results on 3 of WiLDS datasets with additional 2 datasets in the appendix.

**Other Comments Or Suggestions:**

None

**Other Strengths And Weaknesses:**

Strengths:
The paper is clear, well-organized, and thorough. It effectively examines the use of LoRA for in-domain pretraining to improve results, offering a straightforward and useful method. The study includes detailed comparisons with different PEFT techniques and covers multiple varied datasets, greatly enhancing the overall analysis.

Weaknesses:
While providing useful insights, the paper's novelty is somewhat limited as it mainly expands on existing methods. Additionally, it is well-known that in-domain pretraining boosts fine-tuning performance. Hence, the paper could focus more on showing efficient methods for achieving this rather than simply restating the performance benefits (see the "Claims and Evidence" section for more details).

**Questions For Authors:**

None

**Relation To Broader Scientific Literature:**

This paper builds upon MAE, DinoV2 and LoRA. They cite appropriately all the methods that they use and state clearly their contributions.

**Theoretical Claims:**

There are no theoretical claims.

---

> ### Author Rebuttal · Authors · 2025-03-30
>
> Thank you for your valuable comments and for recognizing our thorough experimental validation across baselines and datasets and for providing useful insights into in-domain parameter-efficient pre-training. You may find responses to your questions below:
>
> **Q: Further justification for which blocks to unfreeze for ExPLoRA**
> We agree that block selection is an important component of our method. In section 7, we systematically analyze the properties of patch embeddings from different transformer blocks:
>
> * Blocks that output patch embeddings with low mean/variance eigenvalues inversely correlate with higher linear probing accuracy on predicting patch position
> * Linear probing accuracy for image class peaks in final layers (which also have low mean/variance eigenvalues), while patch position accuracy peaks in middle layers
> * The eigenvalue patterns and classification accuracy suggest unfreezing block L or L-1 would have highest impact for extended pre-training, which is confirmed in section 6.1.2
> * ExPLoRA results in lower mean/variance eigenvalues in the unfrozen block's output, and higher classification and localization accuracies across all ViT layers
> * Linear probing performance in section 6.1.2 follows: block 23 > block 22 > block 9 > block 1, matching exactly the trend in the mean eigenvalue plot (figure 3)
>
> Upon your suggestion, we've included additional baselines with blocks [L-1,L] and [1,L-1,L] unfrozen in the expanded version of table 3 [linked here](https://imgur.com/a/tWdN4xU). We find that increasing unfrozen blocks with ExPLoRA indeed improves linear probing accuracy, where our highest performing D-ExPLoRA model with blocks [1, L-1, L] unfrozen and LoRA-r64 on the rest achieves 78.04% linear probing accuracy, a 9% improvement over prior SoTA.
>
> Note that while more unfrozen blocks improve performance, each adds to the parameter count. We aim to demonstrate strong results with a limited parameter budget. Our analysis in section 7 provides guidelines for selecting additional blocks to unfreeze when higher parameter budgets are available.
>
> Similar trends as in figures 3-7 held across different datasets, guiding our design choices in other experiments.
>
> **Q: Provide a DinoV2 full pre-training baseline for linear probing for fMoW-RGB**
> We agree that this comparison would be useful, however, we want to note that no such baseline in prior literature exists. The SatMAE/ScaleMAE/CrossScaleMAE models were pre-trained on fMoW-RGB, with which we provide direct comparisons to demonstrate that our efficient in-domain pre-training vastly outperforms existing SoTA pre-training methods in the literature by 8% while using 8x-10x less compute.
>
> Upon your suggestion, we have trained a full DinoV2 baseline on fMoW-RGB in the [revised Table 3 linked above](https://imgur.com/a/tWdN4xU). This required ~1200 GPU hours, more than 12x the compute required for ExPLoRA and failed to match its linear probing performance. This likely suggests that much more compute and time is required to pre-train a DinoV2 on fMoW-RGB.
>
> **Q: The paper expands on existing methods**
> With ExPLoRA, we demonstrate a highly effective alternative to expensive full pre training for new image domains. While it combines existing methods, we would like to cite peer conference [NeurIPS guidelines](https://neurips.cc/Conferences/2024/ReviewerGuidelines), which mentions that demonstrating the effectiveness of combining existing techniques can provide substantial research value.
>
> ---
> Thank you again for your time in reviewing our work. We look forward to addressing any follow-up questions you may have. If your concerns are addressed, we kindly ask that you reconsider your score.

---

### Official Review · Reviewer_W3i1 · 2025-03-15

**Overall Recommendation:** 4

**Summary:**

The authors aim to transfer knowledge from large pre-trained vision models to new domains efficiently, and address different downstream tasks. So, given a set of downstream tasks on a new domain, the straightforward approach is to either pre-train from scratch a large model on this new domain and then fine-tune it on the downstream tasks, or to directly fine-tune an existing large model from a different domain on the downstream tasks at hand.

Instead of these approaches, the authors propose Extended Pretraining with LoRA (ExPLoRA), which consists of the following 2 steps. First, a large vision model pre-trained on natural images, like DinoV2 or MAE, is further pre-trained on a new domain, e.g., satellite images. In this pre-training stage there is no use of labels, and the optimization objective is the same as the initial pre-training on the natural images. In this stage, the model in not fully fine-tuned, instead, 1 or 2 layers are fully unfrozen, and LoRA is used for the rest of the layers. In the second step, after the extended pre-training, the model undergoes supervised fine-tuning with a PEFT method on downstream tasks, e.g., classification of satellite images. The core idea is that extended pre-training can be considerably more efficient compared to training from scratch on a new domain, while performance is not compromised, and can even improve, especially compared to directly fine-tuning on downstream tasks.

The authors conduct multiple experiments to validate their method and explore its behavior. They use DinoV2 and MAE as pre-trained models on natural images, and they show that in most cases, ExPLoRA outperforms both models trained from scratch on new domains, as well as models directly fine-tuned on downstream tasks. Their primary experiments are on satellite images, both RGB and spectral, but they show similar results on different domains as well, e.g., WiLDS datasets. Also, they show promising results across different downstream tasks, e.g., image classification, segmentation, and object detection. In addition, they conduct ablation studies to justify their design choices and explore the features learned by ExPLoRA. Finally, additional experiments and analysis of the models behavior is provided in the supplementary material, e.g., the effect of the model size, or that of the duration of extended pre-training.

## update after rebuttal
The authors addressed the main points of my review, so I increased my score from 3 to 4.

**Claims And Evidence:**

- Performance:
    - The authors compare against the 2 main settings they aim to improve on, which is training from scratch on a new domain, and fine-tuning a pre-trained model from a different domain. I think the authors demonstrate clearly that ExPLoRA can lead to performance benefits, since it outperforms or achieves comparable performance across multiple datasets of different size and content, downstream tasks, heads (linear probe and original heads), and backbones (DinoV2 and MAE).
- Efficiency:
    - It is clear that ExPLoRA uses much fewer parameters compared to training a model from scratch, since it unfreezes 1 or 2 layers, and commonly uses LoRA with up to $r=64$ for the rest of the layers. However, the required compute is not included in any of the Tables or Figures. For example, in Table 1, D-[L]-r64 with LoRA-r8 PEFT outperforms DinoV2 with LoRA-r8 by 1.2%, but at what cost in terms of compute? If I am not mistaken, in this case the ExPLoRA model and the baseline undergo the same fine-tuning on the downstream task, but the ExPLoRA model requires an additional pre-training phase, which according to Section C.1. in the Appendix, it corresponds to 200,000 additional training iterations. I understand that it may worth conducting the pre-training if performance is a priority, or if the ExPLoRA model will be used for multiple downstream tasks, so, I want to clarify that I don’t think ExPLoRA should have comparable training GPU hours with all baselines, but I think this should be clearly mentioned in the experiments, since efficiency is a primary motivation for this work.
    - In addition, in Section B.2. in the Appendix, we can see that the DinoV2 baselines reach close to their peak performance after as few as approximately 30 GPU hours, while the best ExPLoRA model needs 420 GPU hours to reach its peak performance. So, I think the authors should have more experiments like this included in the main text (not just in the Appendix), and explicitly discuss compute requirements.

**Essential References Not Discussed:**

Nothing to add.

**Experimental Designs Or Analyses:**

I think the experiments are well designed, and as I mentioned before, the authors do extensive evaluations, covering diverse scenarios. One remark I have is that in most experiments, the backbone is ViT-L, which raises questions about whether the presented results generalize to bigger scales. To their credit, the authors conduct experiments with more diverse backbones in Section B.5. in the Appendix, where they experiment with ViT-G, so, given the importance of model scale for modern applications, I would suggest to include and/or discuss such experiments in the main text.

**Methods And Evaluation Criteria:**

I think the authors use appropriate baselines, datasets and ablation studies. As I mentioned in the previous section, what I think is missing, is an elaborate discussion about compute.

**Other Comments Or Suggestions:**

- Eq. 6: I think notation $\Theta (r)$ is a bit confusing based on the explanation provided “where $r$ controls the fraction of trainable weights”, and the fact that $r \in [0, \infty]$. $r$ is usually used to represent the LoRA rank, which is the way it is used in Eq. 5 as well.
- ln 321, col 1: I think ExPLoRA-L-r8 is in row 11 instead of 10.
- Table 3 is discussed in Section 6.1.2., and there are multiple references to different rows, e.g., rows 11-13. It would be easier to follow such references if the rows of Table 3 were indexed.
- ln 314, col 2: I think it should be “row 3” for LoRA-tuned MAE instead of “row 4”. Similarly, in ln 315, should be “row 7” instead of “row 6”.
- In the iWildcam experiment (paragraph that starts in ln 375, col 2), the authors discuss why the linear probing performance of ExPLoRA suffers in comparison to DinoV2, emphasizing on the small domain gap. However, it is not clear to me why this is a valid hypothesis. In particular, this could explain ExPLoRA and DinoV2 having a small difference in performance, and not DinoV2 clearly outperforming ExPLoRA. I think it would be useful to clarify this argument further.
- In Fig. 3-6, the legend has names “D-ExPLoRA-blk23r64” and “M-ExPLoRA-blk23r64”, I guess “blk23” corresponds to the unfrozen block, but this notation is not used anywhere else in the paper; the same naming is used in Fig. 9 as well.

**Other Strengths And Weaknesses:**

I would like to mention that the manuscript is very well written and easy to follow. Also, the authors motivate well the importance of satellite images, and why they focus on this image domain.

**Questions For Authors:**

I don’t have additional questions.

**Relation To Broader Scientific Literature:**

The proposed method directly combines in a new way 2 fundamental ideas in the literature, unsupervised pre-training (e.g., DinoV2), and PEFT (e.g., LoRA). The main contribution is that PEFT should not be seen solely as a method to address downstream tasks when a large pre-trained model is available, but it is beneficial to first apply PEFT to extend pre-training, and then re-apply PEFT on downstream tasks.

**Theoretical Claims:**

There aren’t any theoretical claims or proofs.

---

> ### Author Rebuttal · Authors · 2025-03-30
>
> Thank you for your detailed and insightful feedback. We appreciate your recognition of ExPLoRA’s novelty in combining fundamental ideas and its strong empirical performance across datasets.
>
> **Q: Discussion of required compute of ExPLoRA vs fine-tuning baselines**
> We agree that more details on compute requirements is needed, which we have added to the main text. Upon reviewing the feedback, we'd like to clarify:
>
> * ExPLoRA is a pre-training method and is meant to provide an efficient alternative to domain-specific _full or continual pre-training_ (eg: SatMAE, ScaleMAE, GFM, etc.). These methods require at least 8x more compute than ExPLoRA pre-training and achieve lower or similar performance to ExPLoRA across different domain gaps (eg: RGB, multi-spectral, temporal data).
> * Directly comparing compute requirements for fine-tuning vs extended pre-training + fine-tuning paints an incomplete picture. The ExPLoRA pre-training phase is functionally different from fine-tuning a natural-image pre-trained model. A pre-trained checkpoint serves multiple purposes- feature extraction, linear-probing, fine-tuning on various downstream tasks etc. In contrast, compute used for supervised fine-tuning yields a model usually suited for just one task (eqs 2, 3). Pre-training compute is amortized across multiple downstream tasks, and would be double counted if included in the fine-tuning budgets for each specific task.
> * We realize now that Figure 7 in section B.2 is incomplete. It only compares the compute for fine-tuning a natural image model (eg: DinoV2) vs. pre-training + fine-tuning for ExPLoRA, for the fMoW-RGB task. Crucially, the pre-training + fine-tuning compute required for full domain-specific pre-training (eg: SatMAE, ScaleMAE) is missing. Further, for different domains (eg: multi-spectral satellite images), the gap between fine-tuning a natural-image model (eg: MAE) and domain-specific pre-training is much larger than for fMoW-RGB (i.e. 6%, from Table 4). Lastly, to reiterate, a directly fine-tuned DinoV2 on fMoW RGB is less flexible than an ExPLoRA pre-trained model, as the latter can be used as any _pre-trained model_- for feature extraction, probing, or further task-specific fine-tuning.
>
> We've expanded Tables 1 and 4 with compute details [linked here](https://imgur.com/a/iMjgOHv). Note that:
> * We only need to allocate 100 GPU hours to ExPLoRA pre-training to achieve the reported gains. For fairness, we use ~220 GPU hours for fine-tuning across models.
> * Providing extra 100 GPU hours to fine-tuning non-ExPLoRA baselines (for an equitable total compute comparison) doesn't improve top-1 accuracy.
> * VPT techniques significantly increase fine-tuning compute by extending token sequence length.
>
> **Q: The DinoV2 baselines reach close to their peak performance in 30 GPU hours while ExPLoRA requires 420 GPU hours**
> This is not a direct comparison, since Figure 7 provides different fine-tuning variants. The red curve is DinoV2 fine-tuned with LoRA-r8, which we also use for ExPLoRA models. The orange and purple curves have blocks unfrozen with LoRA-r64, representing a similar parameter budget to ExPLoRA pre-training. The orange curve reaches peak performance at ~260 GPU hours, by which point all ExPLoRA models have surpassed it.
>
> **Q: Experimental results on larger backbone sizes beyond ViT-L**
> Thank you for the suggestion. As you mention, we compare with ViT-B and ViT-G in Table 15 (appendix B.5). ViT-G experiments are more expensive (>3x parameters vs ViT-L), especially on academic compute budgets. Most prior work uses ViT-L as their backbone, so we maintain this architecture for fair comparison in our main experiments.
>
> **Q: More detailed descriptions of datasets**
> Thank you- we'll include further dataset details in the appendix. For fMoW:
> * fMoW-RGB: 363k training, 53k test data points
> * fMoW-Sentinel: 713k training, 85k test data points
>
> **Q: lines 760-761, parameter budget clarification**
> You're right - it should be 18.7M.
>
> **Q: ln 765: Is it 320 GPU hours or 420?**
> Within 320 total hours (pre-training + fine-tuning), ExPLoRA achieves >1% improvement over the DinoV2 baseline. The 420 hour limit was added as a buffer to demonstrate earlier convergence.
>
> **Q: Eq.6 $\Theta(r)$ notation**
> Thank you- the notation should be $\Theta$ to refer to an unconstrained parameter space (regular full-rank weights).
>
> **Q: iWildcam linear probing result**
> The unfrozen ViT block likely overfit to the training data due to the small domain gap. This effect disappears with PEFT (showing improvement over DinoV2) since all ViT attention layers' Q,V matrices are tuned with LoRA.
>
> **Q: Row number clarifications and "blk" notation in figures 3-6**
> Thank you! We've corrected these mistakes in our manuscript.
>
> ---
> Thank you for your thorough feedback which has improved our paper. Please let us know if you have follow up questions. If your main concerns are resolved, we kindly request that you reconsider your score.

---

> > ### Comment · Reviewer_W3i1 · 2025-04-03
> >
> > I would like to thank the authors for responding to all issues I raised. I have the following comments:
> > - Discussion of required compute of ExPLoRA vs fine-tuning baselines
> >     - I appreciate the detailed answer, I agree with the authors' comments, and the updated Tables are really useful. Though, I would like to ask, why only Table 1 and Table 4, and why the updated Tables are subsets of the original ones? I think compute measurements should be provided for all experiments.
> > - More detailed descriptions of datasets
> >     - I think there are more datasets without details about the split sizes, e.g., WiLDS. I would suggest the authors to go through all the reported datasets and add missing details.
> > - In Appendix C.5. is mentioned that "Hyperparameter and training configuration details are the same as in appendix C.1", does this include number of training iterations? because there are significant differences in the dataset sizes.
> > - One last question, what is the architecture of the heads used with PEFT fine-tuning for downstream tasks? They were the same for the baselines? I apologize in advance if this information is already included, I realized I didn't remember since I first reviewed the paper, and I couldn't find this information by skim reading through the manuscript again.
> >
> > In summary, if the authors can confirm that will add in the updated manuscript a discussion about compute which includes the points they make in "Q: Discussion of required compute of ExPLoRA vs fine-tuning baselines", and assuming that there won't be any issue with my additional questions, especially since the provided updated Tables already address my main concerns, I would be happy to increase my score.

---

> > > ### Author Response · Authors · 2025-04-05
> > >
> > > Thank you again for your detailed review and support of our work. For answers to your follow up questions:
> > >
> > > **Q: Why were only Table 1 and Table 4 updated, and why are they a subset of the original?**
> > > Yes, we agree-- we only provided subsets of Table 1 and 4 for brevity in the rebuttal. We will provide compute details for all experiments in our updated paper, such as [at this link](https://imgur.com/a/4xxC3bO), in tabular format for our main experiments, and in the appendix for other experiments if space does not permit. Thank you for letting us know that the updated tables are useful and that they resolved your main concerns.
> > >
> > > **Q: More detailed description of datasets (eg: WiLDS)**
> > > We will provide full split details for all datasets in the revised appendix for the camera-ready. Here are the train/val splits for all datasets for your reference:
> > >
> > > | Dataset | #Train | #Validation |
> > > |---------|--------|-------------|
> > > | fMoW-RGB | 363.6k | 53.0k |
> > > | fMoW-Sentinel | 712.9k | 84.9k |
> > > | fMoW-Temporal | 83.4k | 14.2k |
> > > | SpaceNet V1 | 6.0k | 1.5k |
> > > | Resisc-45 | 18.9k | 6.3k |
> > > | NAIP | 244.4k | 55.5k |
> > > | EuroSAT | 16.2k | 5.4k |
> > > | Camelyon17 | 302.4k | 33.6k |
> > > | iWildcam | 129.8k | 7.3k |
> > > | Globalwheat | 2.9k | 0.4k |
> > > | VisDA2017 | 152.3k | 55.4k |
> > >
> > > **Q: Is the number of training iterations the same across datasets?**
> > > Thank you for catching this. You are right that some datasets have vastly different numbers of training images. For the smaller datasets, we used ExPLoRA for extended pre-training for a smaller number of iterations. We will make sure to provide these details in the modified appendix. The details are summarized here:
> > > * 200k iterations: fMoW-RGB, Camelyon17, VisDA2017, NAIP
> > > * 150k iterations: iWildcam
> > > * 80k iterations: fMoW-Sentinel, fMoW-Temporal, Globalwheat
> > > * 10k iterations: SpaceNet v1
> > >
> > > Resic-45 and EuroSAT required no extra pre-training, since we used our ExPLoRA pre-trained models from fMoW-RGB or fMoW-Sentinel.
> > >
> > > While fMoW-Sentinel is the largest, processing each multi-spectral input via the group-channel ViT is also more expensive than for an RGB input since the input token sequences are 2-3x longer. We found that 80k pre-training steps was sufficient for ExPLoRA’s performance gains, although more pre-training may benefit the model further.
> > >
> > > While Globalwheat is a small dataset, each image is of a much higher resolution (1024x1024). Thus we are able to pre-train at a 224x224 resolution for 80k steps since data augmentation techniques such as random crops provide enough variety for pre-training.
> > >
> > > **Q: What is the architecture of the heads used with PEFT fine-tuning for downstream tasks?**
> > > The architecture used for PEFT depends on both the task and the PEFT method. We keep a given PEFT architecture the same across all experiments for a given task, unless specified otherwise such as in B.2 Figure 7. For example, for image classification in fMoW-RGB with LoRA-r8 PEFT, the “head” is a trainable linear head that is initialized from scratch. The rest of the ViT backbone is frozen. The trainable LoRA “adapters” are initialized from scratch with the specified rank (eg: 8), and applied on Q, V matrices of all attention layers. For a given PEFT configuration, the main thing that differs is the initialization of the frozen ViT backbone, which can be DinoV2 (or other natural image baselines), ExPLoRA, or fully-pretrained baselines (such as SatMAE, ScaleMAE etc.) For segmentation, we used the same PSANet head architecture used in SatMAE, and for detection, we use Detectron2 as specified in the Globalwheat experiment.
> > >
> > > **Q: Can the authors confirm that the updated manuscript will include a discussion about compute?**
> > > Yes, we can confirm that our updated manuscript will include this discussion and the points we made in our rebuttal response. We thank you again for your insightful feedback, and we also agree that the discussion on compute provides valuable detail to future users of our method- especially as ExPLoRA requires ~10x less compute than full pre-training baselines to achieve the same or higher performance.
> > >
> > > ---
> > > We hope that these explanations resolve your remaining questions. We appreciate your willingness to increase your rating of our work and thank you again for your vote of confidence in our method.

---

### Decision · Program_Chairs · 2025-05-01

**Decision:**

Accept (poster)

**Comment:**

ExpLoRA investigates a new mode of transfer learning: parameter efficient fine-tuning to first extend self-supervised pre-training of some parameters + LoRAs then supervised learning on LoRAs alone for visual data in the presence of shift. The experiments show the approach is effective on remote sensing and standard benchmarks of shift like WILDS. Ablations and analyses provide additional information for practitioners in computer vision, and the results are relevant by their scale with ViTs and by their improvement over full pre-training and fine-tuning.

Reviewers differ in their scores at 3, 3, 2, 2. The authors provide a rebuttal, 4/4 reviewers acknowledge it, and there is extensive discussion. XYuv raises their score from 2 to 3 and W3i1 raises their score from 3 to 4. 83s5 maintains their score for 2 and argues for rejection due to lack of novelty as an application of LoRA to ViT and insufficient efficiency. In the final AC-reviewer discussion 83s5 moderates their view to borderline and XYuv argues in favor of acceptance because the work provides insight for the transfer of foundational models to new visual domains.

The area chair sides with acceptance: the design and thorough experiments are informative and novel in the context of unsupervised then supervised learning. It is not trivial to coordinate the updates of different parameterizations (frozen/unfrozen layers, LoRAs, etc.) and this work offers practical guidance.